# UltraLLaDA: Scaling the Context Length to 128K for Diffusion Large Language Models

**Guangxin He**[1], **Shen Nie**[2], **Fengqi Zhu**[2], **Yuankang Zhao**[3], **Tianyi Bai**[1], **Ran Yan**[1]
**Jie Fu**[4], **Chongxuan Li**[2], **Binhang Yuan**[1†]

[1]HKUST  [2]Renmin University of China
[3]University of Chinese Academy of Sciences  [4]Shanghai AI Lab

## Abstract

Diffusion LLMs have attracted growing interest, with plenty of recent work emphasizing their great potential in various downstream tasks; yet the long-context behavior of diffusion LLMs remains largely uncharted. We present a case study of post-training techniques for extending the context window of diffusion LLMs (i.e., LLaDA) without retraining from scratch. We show that a simple modification to the standard Rotary Positional Embeddings (RoPE) extension effectively accommodates the probabilistic modeling inherent in the diffusion process, enabling stable scaling to longer context ranges. We further compare masking strategies used during post-training and analyze their impact on optimization stability and long-range recall. Instantiating these insights, we introduce UltraLLaDA, a diffusion LLM with a 128K-token context window that, in our empirical evaluation on long-context tasks, significantly outperforms training-free baselines. Our experimental results highlight the special positional extension as a key lever for scaling diffusion LLMs to extended contexts and offer practical guidance for practitioners seeking 128K-scale context via efficient post-training.

## 1 Introduction

Diffusion-based large language models (LLMs) (Gulrajani & Hashimoto, 2023; Lou et al., 2024; Nie et al., 2025) have recently emerged as a promising new paradigm in natural language processing. Unlike popular auto-regressive LLMs (Liu et al., 2024; Dubey et al., 2024; Yang et al., 2025a), which generate text token-by-token, diffusion LLMs employ an iterative denoising process over the entire sequence, offering potential advantages over auto-regressive language models (such as bidirectional global context awareness, flexibility in conditional control, and unified modeling across modalities (Li et al., 2025)). Substantial research has already explored the scalability (Nie et al., 2025), multimodal extensions (Yang et al., 2025b; You et al., 2025), reasoning (Zhu et al., 2025; Zhao et al., 2025), and efficiency optimizations (Ma et al., 2025; Arriola et al.; Wu et al., 2025) of diffusion LLMs. However, one critical aspect remains largely unexplored: *how can we effectively scale and extend the context window of diffusion LLMs beyond their original training limit?* Concretely, the problem is about how to enable diffusion LLMs to handle vastly longer input sequences (e.g., up to 128K tokens) to achieve solid performance on long-context language modeling tasks.

Unlocking long-context capabilities would significantly broaden the applicability of diffusion LLMs. Many real-world language tasks (e.g. processing long documents, multi-turn dialogues, or retrieval-based question answering (Liu et al., 2025a)) demand handling inputs far exceeding the few-thousand-token contexts used in standard training. If the context window of diffusion LLMs could be scaled to such lengths, we could explore if the advantage of diffusion LLM could be further amplified in maintaining coherence and handling complex dependencies across much larger texts than auto-regressive LLMs. Furthermore, preliminary observations suggest diffusion LLMs behave differently with long inputs, e.g., they exhibit stable perplexity under context length extrapolation, whereas auto-regressive LLM typically see perplexity blow up catastrophically once past their trained context limit (Liu et al., 2025b). To move forward, we believe it is crucial to fully realize the potential of diffusion LLMs in long context scaling to systematically compare their performances with auto-regressive LLMs in tasks requiring extensive long context.

---

[†]Corresponding to: Binhang Yuan (`biyuan@ust.hk`).

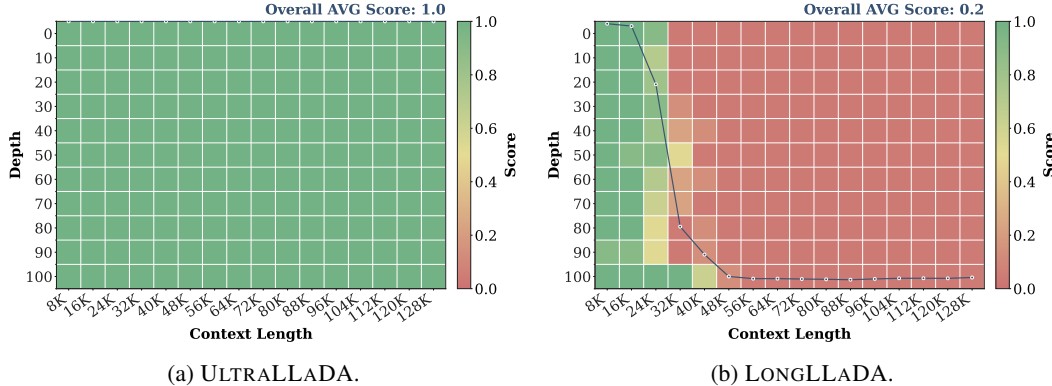

(a) ULTRALLADA.                                    (b) LONGLLADA.

Figure 1: NIAH evaluation up to 128K context-length. ULTRALLADA can find all of the needles within the context window 8–32× longer than that LONGLLADA can handle.

Achieving this goal, however, is non-trivial. Prior research in scaling the context window of auto-regressive LLM indicates that naively increasing the position range of the model (for example, by extrapolating Rotary Positional Embeddings (Chen et al., 2023) beyond the length seen in training) can lead to mismatches with the model's learned positional dynamics and generation process. Such a performance degradation could be inherited from the attention patterns learned on shorter contexts, which are not able to directly accommodate the extended positional embeddings. Recent attempts of scaling diffusion LLMs have further reveals some interesting observations, e.g., diffusion LLMs do not suffer a total collapse in perplexity when input length grows (unlike auto-regressive LLMs), but they instead display a "local perception" bias, which means that when a diffusion LLM is given a context longer than it was trained on, it tends to only utilize information from the most recent segment (e.g. the last 4K tokens) and ignore the content that lies far back in the prompt (Liu et al., 2025b). However, such observation was derived from a *training-free* method (Liu et al., 2025b). Based on the prior research on auto-regressive LLMs, *post-training* based methods for extending the context length of LLMs are often superior to training-free methods because they allow the model to fundamentally adapt its internal mechanisms to handle longer sequences (Liu et al., 2025a).

In this paper, we plan to explore how to effectively scale and extend the context window of diffusion LLMs by answering the following two concrete questions:

- **Q1.** *How can we effectively scale and extend the context window of diffusion LLMs derived from the existing methods originally designed for auto-regressive language models?*

- **Q2.** *What is the corresponding potential performance boost when comparing the post-training based methods for diffusion LLMs with the training-free method (i.e., LongLLaDA)?*

**Contribution 1.** To answer the first question, we introduce a diffusion-specific extension of Rotary Positional Embeddings that enables better long-context modeling. In particular, we develop a Diffusion-aware NTK method — an adaptation of the Neural Tangent Kernel method tailored to diffusion LLMs — that accommodates the iterative denoising process and permits RoPE extrapolation to 128K tokens. We also investigate different masking strategies for light-weight post-training on extended contexts (including adaptive masking and end-of-document concatenation) and analyze their impact on optimization stability and long-range information recall. These methodological contributions provide a principled foundation for scaling the context window of diffusion-based language models in a post-training setting, where we extend the context length of one state-of-the-art diffusion LLM, i.e., LLADA (Nie et al., 2025) to up to 128K tokens, namely ULTRALLADA.

**Contribution 2.** To answer the second question, we present comprehensive experiments validating the effectiveness of ULTRALLADA across multiple long-context benchmarks. We compare ULTRALLADA against LONGLLADA and the original LLADA base model, observing that ULTRALLADA consistently outperforms these alternatives when handling extremely large context (e.g., Figure 1). On a diverse benchmark suite of long-context tasks, ULTRALLADA achieves superior performance, maintaining low perplexity and high task accuracy as context length increases. This extensive empirical evaluation highlights ULTRALLADA's state-of-the-art long-context capabilities and demonstrates the practical benefits of our lightweight post-training approach.

## 2 PRELIMINARY AND RELATED WORK

**Diffusion Language Model.** Diffusion models have recently emerged as a promising alternative to the auto-regressive models for language generation (Li et al., 2025). Unlike AR LLMs that predict tokens sequentially, diffusion language models generate text through an iterative denoising process conditioned on noised inputs. Among these methods, one popular paradigm known as masked diffusion language model (Austin et al., 2021; Lou et al., 2023; Sahoo et al., 2024; Shi et al., 2024; Ou et al., 2024) utilizes a forward process to incrementally introduce noise into discrete data, and a learned reverse process to generate coherent sentences. Given a noise level $t \in [0, 1]$ and input data $\boldsymbol{x}_0 \in \{0, 1, \ldots, K-1\}^L$, where $K$ is the vocabulary size and $L$ is the sequence length, the forward process $q(\boldsymbol{x}_t|\boldsymbol{x}_0)$ independently replaces each token in $\boldsymbol{x}_0$ with a mask token with probability $t$, and leaves it unchanged with probability $1 - t$. Masked diffusion LLMs are usually trained to minimize the upper bound on the negative log-likelihood (Sahoo et al., 2024; Shi et al., 2024; Ou et al., 2024):

$$-\mathbb{E}_{t \sim U[0,1], \boldsymbol{x}_0 \sim p_{\text{data}}, \boldsymbol{x}_t \sim q(\boldsymbol{x}_t|\boldsymbol{x}_0)} \left[ \sum_{\{i|\boldsymbol{x}_t^i = m\}} \log p_{\boldsymbol{\theta}}(\boldsymbol{x}_0^i|\boldsymbol{x}_t) \right], \tag{1}$$

where $m$ denotes the mask token, and $p_{\boldsymbol{\theta}}$ is parameterized by a bidirectional Transformer (Vaswani et al., 2017). During inference, masked diffusion language models denoise starting from the concatenation of a prompt and a masked sequence. At each denoising step, $p_{\boldsymbol{\theta}}$ predicts the original tokens for all masked positions. Afterward, a subset of the predicted tokens is re-masked, and the number of tokens to be re-masked is treated as a hyperparameter that controls the trade-off between generation quality and decoding speed. LLaDA (Nie et al., 2025) and Dream (Ye et al., 2025) are two of the representative open-source diffusion language models. LLaDA is trained from scratch, while Dream is obtained by post-training a pre-trained auto-regressive language model. Both models achieve performance that is competitive with strong LLMs such as LLaMA3-8B (Dubey et al., 2024) and Qwen2.5-7B (Qwen et al., 2025).

**Extending the Context Window of LLMs**. Extending the context length of LLMs has become crucial for tasks such as long document processing and multi-turn conversations. Recent research has developed a range of techniques to extend context windows far beyond the few thousand tokens during pre-training (Liu et al., 2025a). In state-of-the-art LLM transformer architectures (e.g., LLaMA (Touvron et al., 2023)), rotary position embedding (RoPE) (Su et al., 2024) is a popular position embedding method, which enhances traditional position embeddings by encoding positions using rotating vectors in the complex plane and preserves relative positional relationships, enabling better generalization to extended contexts. Various methods have been proposed to extend the context window: Position interpolation (PI) (Chen et al., 2023) generates positional embeddings for out-of-training-range positions by interpolating between embeddings of nearby positions; NTK-Aware Scaling (Peng & Quesnelle, 2023) rescales the RoPE base to preserve the neural tangent kernel (NTK)—a linear approximation of the model's behavior—when extending the context; YaRN (Peng et al.) extends RoPE by dynamically scaling rotation frequencies based on layer depth and target context length to maintain rotational consistency across layers and contexts. Noted that most of the current context extending methods are based on auto-regressive LLMs — only a very recent research, i.e., LongLLaDA (Liu et al., 2025b), has initialized the research for diffusion LLMs, which integrates an NTK-based RoPE extrapolation into a diffusion LLM based on a training-free paradigm. To make an apple-to-apple comparison, our training-based extension is also based on NTK, and we summarize its formulation below (Peng & Quesnelle, 2023):

$$\lambda_{NTK} = s^{\frac{d}{d-2}}, s = \frac{T_{\text{target}}}{T_{\text{train}}}. \tag{2}$$

Here $d$ denotes the (even) rotary dimension, $T_{\text{train}}$ and $T_{\text{target}}$ represent the pretraining context length and desired extended context length; $\lambda_{\text{NTK}}$ is the global scaling factor applied to the RoPE base. Note that NTK scaling multiplies the RoPE base by the same factor $\lambda$ across all rotary dimensions.

## 3 LONG-CONTEXT SCALING FOR DIFFUSION LLM

To answer **Q1** (*how to scale diffusion LLM context windows*), we first revisit NTK-based RoPE extrapolation from LONGLLADA (Liu et al., 2025b) in §3.1; propose a new diffusion-aware NTK extension approach for ULTRALLADA in §3.2; and enumerate the details of our strategies for lightweight post-training on long-context data that mitigate cross-segment noise in §3.3.

## 3.1 REVISIT NTK-BASED ROPE EXTRAPOLATION IN LONGLLADA

LONGLLADA (Liu et al., 2025b) applied NTK-aware RoPE scaling originally designed for auto-regressive LLMs to diffusion LLMs, demonstrating that simple RoPE base scaling can extend a diffusion model's context length in a training-free manner. Concretely, given the RoPE base $b$, rotary dimension $d$, pretraining context length $T_{\text{train}}$, and target context length $T_{\text{target}}$, LONGLLADA first computes a *critical dimension* $d_{\text{crit}}$ based on the LLM's pre-training context length $T_{\text{train}}$, then sets a scaling factor $\lambda_{\text{baseline}}$ via:

$$\lambda_{\text{baseline}} = b^{-1} \cdot \left(\frac{T_{\text{target}}}{2\pi}\right)^{\frac{d}{d_{\text{crit}}}}, \quad d_{\text{crit}} = 2\lceil \frac{d}{2}\log_b \frac{T_{\text{train}}}{2\pi}\rceil \quad (3)$$

**Key Observation and Discussion.** Note that LONGLLADA utilizes the same $T_{\text{train}}$ in context window extension as auto-regressive LLMs, although diffusion (with bidirectional attention) and auto-regressive language models perceive different effective relative span during pre-training. As a result, directly applying auto-regressive-oriented assumptions to diffusion LLMs therefore, misestimates both the critical dimension and the scaling factor. In other words, such an approach inherits extrapolation strategies from auto-regressive LMs without accounting for diffusion-specific properties (notably, *bidirectional attention over the entire sequence*). Consequently, *long-context potential inherited from diffusion language modeling is not fully unlocked.* This motivates us to propose a diffusion-specific NTK extrapolation method (Diffusion-aware NTK) that explicitly incorporates diffusion LLM characteristics.

## 3.2 DIFFUSION-AWARE NTK IN ULTRALLADA

Motivated by the observed gap discussed in §3.1, we introduce *diffusion-aware NTK*, which refines the mapping from $T_{\text{train}}$ to critical dimension by accounting for diffusion-specific properties, yielding an adjusted scaling factor $\lambda^{'}$. Comprehensively, our goal is to better exploit the properties of diffusion LLM's properties to unlock their long-context capability. We define: $T_{\text{cap}}$ as the maximum relative span learned during pretraining on $T_{\text{target}}$ length data, and $T_{\text{Ecap}}$ as the required relative span after context-length extension to $T_{\text{target}}$.

**Diffusion-aware NTK**. We first define a new *scaling factor* $\lambda'$ for RoPE based on an NTK-aware criterion tailored to diffusion LLMs. Concretely, let $d$ be the model's attention dimension and $b$ the original RoPE base. We define the *modified critical dimension* $d'_{\text{crit}}$ as the largest even-indexed dimension whose RoPE sinusoidal period $L_{d_{\text{crit}}}$ does not exceed $T_{\text{cap}}$. Formally, we have:

$$\lambda^{'} = b^{-1} \cdot \left(\frac{T_{\text{Ecap}}}{2\pi}\right)^{\frac{d}{d_{\text{crit}}}}, \quad d'_{\text{crit}} = 2\lceil\frac{d}{2}\log_b \frac{T_{\text{cap}}}{2\pi}\rceil, \quad (4)$$

By reducing the angular frequency, this scaling mechanism increases the RoPE periods across all dimensions. In other words, we choose $\lambda'$ in Equation 4 in this way, so that we can effectively slow down the RoPE rotations, and lengthen the positional wavelengths along all attention dimensions. In particular, the formerly "well-trained" critical dimension $d'_{\text{crit}}$ now has an expanded period covering $T_{E\text{cap}}$, thereby unlocking the model's ability to attend over the extended context range.

It is crucial to note that diffusion LLMs have a different effective $T_{\text{cap}}$ from auto-regressive LLMs. In a diffusion model with bidirectional attention, each token attends to the entire sequence in both directions, so during training it effectively learns relative positions in the range $[-(T_{\text{train}}-1), T_{\text{train}}-1]$. This symmetric coverage roughly doubles the span of learned positions (i.e., $2T_{\text{train}}$) compared to an auto-regressive LLM, which only attends to past tokens (relative positions $[0, T_{\text{train}} - 1]$). As a result, a diffusion LLM can naturally handle a wider range of relative positions(see Appendix C for the theoretical analysis). We account for this by setting $T_{\text{cap}} \approx 2T_{\text{train}}$ and $T_{\text{Ecap}} \approx 2T_{\text{target}}$ (versus $T_{\text{cap}} \approx T_{\text{train}}$ and $T_{\text{Ecap}} \approx T_{\text{target}}$ for auto-regressive models) when computing $\lambda$. In practice, using this diffusion-aware $T_{\text{cap}}$ leads to a slightly larger critical dimension and a more conservative (smaller) $\lambda'$ than the original NTK formula, i.e., $\lambda_{\text{baseline}}$ in Equation 3. For example, as illustrated in Figure 2a, for the 8B LLADA model with $T_{\text{train}} = 4K$, the baseline auto-regressive centric formula yields $d_{\text{crit}} \approx 64$ (period ~4K tokens), whereas our diffusion-aware approach gives $d'_{\text{crit}} \approx 70$ (period ~8K tokens). For simplicity, we conduct training-free experiments to verify the effectiveness of this method. The preliminary results (Figure 2c, 2b) and this formulation suggests that incorporating bidirectional coverage is essential to extend diffusion LLMs' context properly. A finer-grained sweep over nearby critical dimensions is provided in Appendix D. We adopt this Diffusion-aware NTK scaling for all extended-context experiments in the following sections.

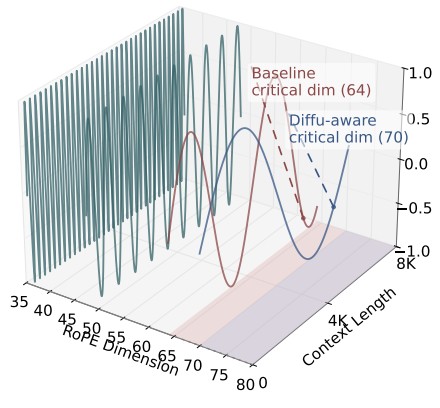

| NTK | NIAH(32K) | LB(16K) |
|---|---|---|
| Diffu-aware | 79.36 | 35.38 |
| Baseline | 75.28 | 35.02 |

(b) Training-free NIAH and LongBench scores.

(a) RoPE critical dimension.

(c) Training-free PPL on 32K context length.

Figure 2: RoPE critical dimension and training-free case study under different NTK scaling.

### 3.3 CASE STUDY OF MASKING FOR DIFFUSION LLM CONTEXT EXTENSION.

After re-scaling the positional embedding using the above method, we further post-train the diffusion LLM on long-context data under the new encoding. We generate long input sequences from the PG19 corpus (Rae et al., 2019) following the packing strategy of (Peng & Quesnelle, 2023): shorter documents are concatenated together to reach a target length (we use 64K tokens per packed sequence), while very long documents are split into consecutive 64K segments (carrying any remainder to the next segment). This packing creates training sequences up to 64K tokens long without altering the overall token distribution.

A key challenge in long-context post-training is dealing with *cross-document interference*. When multiple unrelated texts are concatenated in a single sequence, a vanilla diffusion LLM (with global bidirectional attention) may attend across the document boundaries, causing tokens to mistakenly incorporate context from other documents. Auto-regressive LLMs also face interference when packing data, but their strictly causal (unidirectional) attention (Figure 3a) naturally limits some interaction between unrelated segments. In the diffusion paradigm, because any token can interact with any other token, we need explicit strategies to prevent spurious interactions between unrelated segments.

**Masking in Diffusion LM Long Context Extension**. Previous works on long-context training for auto-regressive LLMs have proposed several approaches: e.g., pack data to a certain length regardless of the document boundary (Fu et al., 2024; Touvron et al., 2023; Raffel et al., 2020), inserting special boundary tokens to mark segment ends (Peng et al., 2023), or using specialized attention masks to block inter-document attention (Gao et al., 2024; Shang et al., 2025). However, the effectiveness of these strategies in diffusion LLMs (with fully bidirectional attention) remained under-explored, motivating us to compare them in our setting. We consider three post-training strategies of masking (illustrated in Figure 3):

- (**i**) *Adaptive attention masking*: We construct a document-aware attention mask for each packed training sequence that allows full attention only within each original document. Any attention between tokens belonging to different source documents is masked out (set to 0). This effectively blocks cross-document influence while preserving bidirectional attention within each document (Figure 3-b schematically shows this masking for three segments). During post-training, the model learns under this segmented attention pattern, ensuring it does not rely on unrelated context.

- (**ii**) *End-of-document (EOD) concatenation*: We insert a special end-of-document token between documents when packing them. The model is still trained with standard full bidirectional attention over the entire sequence (no mask beyond the usual ones), but the EOD token provides a learnable boundary indicator (Figure 3-c). The expectation is that the model will learn to treat EOD as a separator and avoid blending information across it. Unlike Adaptive Masking, this method does not explicitly forbid cross-segment attention, but rather gives the model a chance to infer document breaks from the EOD symbol.

- (**iii**) *Direct concatenation*: As a baseline, we also include naively packed sequences with no special handling, i.e., documents are concatenated back-to-back and the model uses full bidirectional

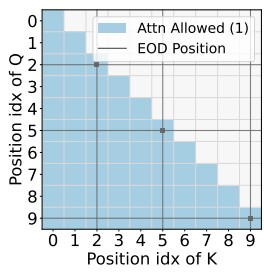 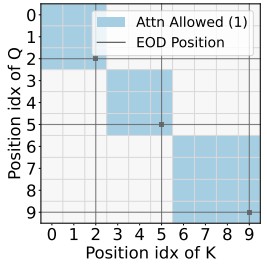 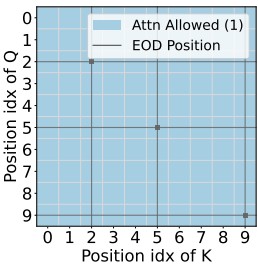

(a) Auto-regressive Attention.  (b) Adaptive Attention (Diffusion).  (c) Full Attention (Diffusion).

Figure 3: Different attention mechanism for Long-context training.

Table 1: Perplexity on 128K long test data.

| PPL | 4K | 8K | 16K | 32K | 48K | 64K | 96K | 128K |
|---|---|---|---|---|---|---|---|---|
| LLaDA-8B-Base | 12.00 | 11.98 | 13.66 | 16.80 | 28.04 | 53.92 | 162.62 | 343.88 |
| LongLLaDA | 13.39 | 13.38 | 14.66 | 16.41 | N/A | N/A | N/A | N/A |
| UltraLLaDA | 11.27 | 11.15 | 11.66 | 11.32 | 11.96 | 11.62 | 11.09 | 10.45 |

attention over the entire sequence with no boundary tokens. This is the simplest approach and may be prone to maximal cross-document interference.

We trained separate long-context models with each of the above strategies (all of them also using our Diffusion-aware NTK position scaling). *Empirically, both adaptive masking and EOD concatenation dramatically reduce cross-document interference compared to the direct concatenation baseline.* The model post-trained with direct concatenation often produces incoherent results, presumably due to unrelated content bleeding together. In contrast, both alternative strategies yield much more coherent generations, with Adaptive Masking showing a slight advantage over EOD tokens in our experiments. By either explicitly preventing cross-document attention or clearly marking the boundaries, these approaches allow the diffusion LLM to fully exploit its long-context capacity during post-training. Overall, our combined approach — applying Diffusion-aware NTK RoPE scaling followed by long-context post-training with an appropriate long sequence data processing strategy (adaptive masking or EOD markers) — substantially improves the model's ability to handle extended context windows. As we will show next, the resulting model (namely UltraLLaDA) maintains coherent, low-perplexity performance even far beyond its original training context.

## 4 EXPERIMENTS

We thoroughly evaluate the long-context capabilities of UltraLLaDA and compare it against baseline models to answer **Q2** (*the performance gain of our post-training approach over training-free method*), and against different post-training settings to further discuss **Q1** (*how to effectively scale and extend the context window of diffusion LLMs*). Experiments are conducted on four benchmarks that stress-test long-context capabilities of UltraLLaDA up to 128K tokens:

- **PPL-128K**: Language modeling perplexity evaluation on a 128K-token test document from PG19. Lower perplexity better predictive modeling of long text. We estimate PPL via Monte-Carlo denoising likelihood on randomly masked tokens (cf. Eq. 1), the same method used in LONGLLADA; while lower still indicates better performance, this metric is not strictly identical to auto-regressive language models' next-token PPL.
- **NIAH-128K:** Needle-in-a-haystack long-context retrieval task (gkamradt, 2023), where a single relevant sentence is embedded in a long distractor context (up to 128K tokens) and the model must retrieve it. We report retrieval accuracy using an inference configuration with output length set to 32, block size of 32, and 32 sampling steps.
- **LongBench-16K:** A suite of diverse long-context tasks (Bai et al., 2023) including question answering, summarization, code completion and other tasks, at a context length of 16K. We report the aggregated score (weighted by question count) across all sub-tasks using an inference configuration with output length set to 512, block size of 64, and 512 sampling steps.

Table 2: LongBench cut at 16K Evaluation. Sub tasks: single-document QA (SD), multi-document QA (MD), summarization (Sum), in-context learning (ICL), synthetic tasks (Syn), and code completion (Code). AVG is an aggregated question-count–weighted average score.

| Model | AVG | SD | MD | Sum | ICL | Syn | Code |
|---|---|---|---|---|---|---|---|
| LLaDA-8B-Base | 31.56 | 13.40 | 8.04 | 21.71 | 61.83 | 30.38 | 56.41 |
| LongLLada | 36.07 | 15.14 | 11.26 | 24.77 | 67.22 | 40.56 | 61.54 |
| UltraLLaDA (Ours) | **39.98** | 18.84 | 14.99 | 27.58 | 70.59 | 50.44 | 63.57 |

Table 3: RULER with context lengths 4K to 32K. include Retrieval: Needle-in-a-haystack (NIAH), Aggregation: frequent words extraction (AGG), question answering (QA), and Multi-hop Tracing: variable tracking (VT). AVG is question-count–weighted average score. "–" indicates failure.

| Model | 4K | | | | | 8K | | | | |
|---|---|---|---|---|---|---|---|---|---|---|
| | AVG | NIAH | AGG | QA | VT | AVG | NIAH | AGG | QA | VT |
| LLaDA-8B-Base | 86.17 | 99.19 | 45.84 | 67.50 | 100 | 41.69 | 46.66 | 42.84 | 23.50 | 36.00 |
| LongLLaDA | 87.41 | 97.84 | 73.82 | 53.00 | 100 | 65.20 | 76.28 | 41.20 | 49.50 | 56.00 |
| UltraLLaDA (Ours) | **88.37** | 97.31 | 66.69 | 68.50 | 100 | **86.22** | 98.28 | 55.29 | 62.00 | 100 |
| | 16K | | | | | 32K | | | | |
| LLaDA-8B-Base | – | – | – | – | – | – | – | – | – | – |
| LongLLaDA | 45.48 | 52.75 | 17.70 | 40.00 | 53.80 | 5.69 | 3.69 | 5.45 | 15.50 | 2.60 |
| UltraLLaDA (Ours) | **77.51** | 93.00 | 43.12 | 39.50 | 98.40 | **73.63** | 92.78 | 29.29 | 29.00 | 98.40 |

- **RULER-32K:** A benchmark for long-context models (Hsieh et al., 2024) includes categories like retrieval, aggregation (e.g. common word extraction), question answering, and multi-hop tracing (variable tracking). We evaluate with context lengths up to 32K. Evaluations are conducted with context lengths up to 32K, and we report both the question-count–weighted average score and the breakdown by category. Results are obtained using an inference configuration with output length set to 64, block size of 64, and 64 sampling steps.

For all evaluations, lower is better for perplexity, while higher is better for accuracy scores. We emphasize fairness in comparison: all models use the same base 8B LLaDA initialization and were evaluated under identical conditions. ULTRALLaDA was obtained via a lightweight post-training of 600 steps on long data, whereas the baseline LONGLLaDA model uses a training-free RoPE extrapolation approach without additional training. In all cases we use deterministic decoding to eliminate sampling variance, so we do not report confidence intervals or significance tests. For reproducibility, the training details and model hyperparameters are provided in Table 7 (Appendix B).

## 4.1 MAIN RESULTS.

**NIAH Retrieval.** Figure 1 compares our ULTRALLaDA to the LONGLLaDA baseline on the NIAH retrieval task for context lengths from 4K up to 128K tokens. ULTRALLaDA achieves a 100% retrieval accuracy at every evaluated context length, all the way to 128K. In contrast, the baseline LONGLLaDA (which extended a 4K-trained model to 32K in a training-free manner) performs reasonably at shorter lengths (over 80% accuracy at 8K and 16K) but then drops sharply to around 20% at 32K and fails entirely beyond 32K. In fact, LONGLLaDA could not be evaluated past 32K due to its method's limitations. These results demonstrate that our post-training approach preserves excellent retrieval capability even with extremely long contexts (128K), whereas the training-free baseline rapidly degrades as context length increases. ULTRALLaDA can successfully retrieve the "needle" in contexts that are 8–32× longer than those the baseline LONGLLaDA can handle.

**PPL.** We next assess generative modeling quality on a long document. We took a 128K-token test text from PG19 and measured perplexity when the model is given increasing portions of context (4K up to 128K). Table 1 reports the perplexity of the base model LLaDA, the LONGLLaDA baseline, and ULTRALLaDA. The base LLaDA model (trained on 4K context) experiences a perplexity increase from around 12 at 4K to around 344 at 128K, indicating it cannot maintain coherence beyond

its training length. The LONGLLADA baseline performs even worse at shorter ranges — notably, at 4K and 16K its perplexity is higher (worse) than the base model's — and achieves only a slight improvement at 32K (16.4 vs base 16.8). Furthermore, LONGLLADA cannot be extended beyond 32K (entries marked "N/A" in Table 1). In contrast, ULTRALLADA maintains a low and stable perplexity (11-12) across all lengths up to 128K. This highlights the robustness of our approach in modeling extremely long sequences.

**LongBench.** We evaluate models on the LongBench benchmark truncated to a 16K context window (since many tasks in LongBench do not require more than 16K). Table 2 shows the aggregated scores (higher is better) and breakdowns across six representative sub-tasks: single- and multi-document QA (SD, MD), summarization (Sum), in-context learning (ICL), synthetic reasoning tasks (Syn), and code completion (Code). ULTRALLADA achieves the highest score on every sub-task, outperforming both the base model and the LONGLLADA baseline. Overall, ULTRALLADA's average score is 39.98, a solid improvement over the baseline's 35.38 and the base model's 30.89. This demonstrates that our long-context training not only extends the context length but also yields quality gains on challenging tasks even at 16K (within the baseline's range). We attribute this to improved long-range coherence and understanding gained through our post-training procedures.

**RULER.** Table 3 reports results on the RULER benchmark with context lengths from 4K to 32K (covering retrieval, aggregation, QA, and multi-hop variable tracking (VT) tasks). ULTRALLADA again consistently outperforms both the base model and LONGLLADA at all lengths. The performance gap widens as the sequence length increases. At 8K, the baseline already lags behind (average score 65.20 vs our 86.22). By 16K the baseline's average drops to 45.48 while ours remains 77.51. At the maximum 32K length, the baseline collapses to an average of 5.69, essentially failing on most tasks, whereas ULTRALLADA still achieves 73.63 average. Notably, prior work noted that diffusion LLMs struggled on the VT tasks at long lengths (Liu et al., 2025b). We observe the same for the baseline: its VT score falls to 2.6 at 32K. In contrast, ULTRALLADA maintains a near-perfect VT score of 98.4 at 32K, on par with its performance at 8K. In fact, ULTRALLADA exhibits strong scaling on both the retrieval (NIAH) and tracing (VT) categories, where it achieves 90–100% accuracy across the board. The gains on aggregation (AGG) and some QA tasks are more modest. This suggests that diffusion LLMs, even when scaled to extreme context lengths, particularly excel at tasks requiring pinpoint retrieval of information and maintaining consistency (tracing variables) over long texts, while tasks that involve combining or reasoning over many pieces of information (aggregation, complex QA) remain challenging. Overall, the RULER results reinforce that our post-training method yields substantial performance boosts over the training-free baseline, especially at the longer end of the context range, answering **Q2**.

Beyond the main results reported here, we further provide additional long-context evaluations on more complex comprehension and reasoning benchmarks at longer context lengths in Appendix H. At the same time, similar to prior observations in autoregressive LLMs, extending the context window may introduce a trade-off between long-context gains and short-context performance; we provide additional analysis of this short-context regression in Appendix I.

### 4.2 ABLATION STUDY: DIFFUSION-AWARE VS. BASELINE NTK SCALING.

We perform an ablation study to isolate the impact of our Diffusion-aware NTK scaling (Section 3.2) compared to Baseline NTK scaling used by LONGLLADA. In this experiment, we post-trained two variant models on long data: one using our Diffusion-aware $\lambda'$ calculation and one using the Baseline NTK formula (all other training settings held equal, including the use of the EOD concatenation strategy for data packing). We then evaluated both models on LongBench and RULER tasks. The results are summarized in the first two rows of Table 4 and 5. On the LongBench: Diffusion-aware NTK yields a small but consistent gain in overall score: 39.80 vs. 39.44. On the RULER: At 4K, the baseline variant is marginally higher (90.00 vs. 87.86). From 8K onward, diffusion-aware scaling takes the lead and the margin widens with length: 86.30 vs. 85.30 at 8K, 82.99 vs. 79.54 at 16K, and 70.78 vs. 65.85 at 32K. This trend supports our premise: explicitly accounting for diffusion's bidirectional attention in RoPE scaling better preserves performance as sequences grow. In sum, the gains observed with ULTRALLADA stem not only from additional training data but also from the positional encoding adaptation introduced by Diffusion-aware NTK. Appendix E provides additional long-context experiments comparing diffusion-aware NTK with the baseline NTK under identical post-training settings, showing the advantage of diffusion-aware NTK.

Table 4: LongBench results: Diffusion-aware NTK and mitigating cross-document interference.

| Model | AVG | SD | MD | Sum | ICL | Syn | Code |
|---|---|---|---|---|---|---|---|
| Base-NTK + EOD-Cat | 39.44 | 17.39 | 14.76 | 26.87 | 70.27 | 49.48 | 63.79 |
| Diffu-NTK + EOD-Cat | 39.80 | 17.83 | 13.43 | 27.91 | 69.38 | 51.53 | 64.50 |
| Diffu-NTK + Adapt-Mask | **39.98** | 18.84 | 14.99 | 27.58 | 70.59 | 50.44 | 63.57 |
| Diffu-NTK + Direct-Cat | 38.77 | 17.29 | 13.22 | 27.54 | 70.22 | 50.07 | 60.92 |

Table 5: RULER results: Diffusion-aware NTK and mitigating cross-document interference.

| Model | 4K | | | | | 8K | | | | |
|---|---|---|---|---|---|---|---|---|---|---|
| | AVG | NIAH | AGG | QA | VT | AVG | NIAH | AGG | QA | VT |
| Base-NTK + EOD-Cat | **90.00** | 99.41 | 65.40 | 72.00 | 100 | 85.30 | 97.63 | 52.70 | 62.00 | 99.2 |
| Diffu-NTK + EOD-Cat | 87.86 | 96.56 | 63.22 | 72.00 | 99.20 | **86.30** | 99.22 | 51.70 | 63.60 | 97.60 |
| Diffu-NTK + Adapt-Mask | 88.37 | 97.31 | 66.69 | 68.50 | 100 | 86.22 | 98.28 | 55.29 | 62.00 | 100 |
| Diffu-NTK + Direct-Cat | 89.44 | 98.09 | 65.40 | 74.00 | 99.20 | 85.98 | 97.19 | 55.74 | 66.50 | 95.80 |
| | 16K | | | | | 32K | | | | |
| Base-NTK + EOD-Cat | 79.54 | 93.63 | 46.90 | 46.00 | 99.20 | 65.85 | 76.34 | 27.47 | 46.00 | 98.40 |
| Diffu-NTK + EOD-Cat | **82.99** | 97.44 | 42.49 | 58.00 | 98.40 | 70.78 | 86.47 | 28.60 | 36.00 | 99.20 |
| Diffu-NTK + Adapt-Mask | 77.51 | 93.00 | 43.12 | 39.5 | 98.4 | **73.63** | 92.78 | 29.29 | 29.00 | 98.40 |
| Diffu-NTK + Direct-Cat | 74.49 | 87.66 | 43.07 | 42.00 | 97.00 | 63.04 | 75.72 | 28.39 | 31.00 | 95.00 |

## 4.3 CASE STUDY: MITIGATING CROSS-DOCUMENT INTERFERENCE.

Our second design study examines the importance of the long-context training strategy by comparing the three long sequence data processing approaches from Section 3.3: Adaptive attention masking, EOD concatenation and Direct concatenation. We trained three diffusion LLMs on 64K-context data, all using the Diffusion-aware NTK scaling but each with one of the different document processing strategies. We then evaluated each model on LongBench and RULER to see how the post-training method impacts performance. The results, shown in the last three rows of Table 4 and 5, reveal that explicitly addressing cross-document interactions is crucial for long-context training. On LongBench, both boundary-aware strategies outperform naive packing: Adaptive Masking attains the best overall score, followed by EOD concatenation, while Direct concatenation lags. On Ruler, the benefits of handling document boundaries grow with context length. At 8K, the EOD concatenation is best, narrowly ahead of Adaptive Masking and Direct concatenation. At 16K, the gap widens in favor of EOD concatenation over Adaptive Masking and Direct Concatenation. Crucially, at 32K, Adaptive Masking becomes clearly superior, surpassing EOD Concatenation and far outpacing Direct concatenation. Although Direct concatenation scores slightly higher at 4K than the boundary-aware variants, its advantage vanishes and then reverses as length increases.

**Takeaway.** Between the two boundary-aware strategies, we observe some trade-offs. The EOD concatenation strategy tends to perform better at shorter or moderate lengths. When the context is not extremely long, simply providing boundary tokens is sufficient and perhaps allows the model a bit more flexibility (since it can still attend globally). However, at longer length (e.g. 32K), the Adaptive Masking overtakes EOD concatenation. This indicates that for very long sequences with many concatenated documents, completely blocking cross-document attention yields the most robust results – likely because it prevents any chance of confusion between unrelated content. Crucially, both strategies beat the Direct concatenation. This underscores the importance of explicitly handling document boundaries in long-context training for diffusion LLMs. In summary, our design studies confirm that both core components of our approach – the Diffusion-aware NTK scaling and the improved post-training data strategy – are necessary and effective for scaling diffusion LLMs to long context, answering both research questions **Q1** and **Q2** with a resounding positive result.

## 4.4 GENERALIZATION TO DREAM

To examine whether our training-based scaling strategy generalizes beyond LLaDA, we further apply it to Dream, a representative diffusion LLM converted from an autoregressive model. Dream na-

tively supports a context length of 2K and retains an AR-style shifted-label training objective, which is structurally different from LLaDA's unshifted diffusion training objective. We extend Dream from 2K to 32K using our training-based pipeline and compare three variants: Dream, the original 2K model; Dream-NTK, the original Dream model with direct NTK scaling applied at inference time without post-training; and Dream-Finetune, the 32K Dream model obtained by post-training Dream with our long-context training pipeline. We evaluate these variants on NIAH-32K and LongBench-v2-Short (tasks with context lengths ≤ 32K). The evaluation configuration and prompt template are the same as those used in Appendix E.

Table 6: Generalization of our training-based scaling strategy to Dream.

| Model | N-4K | N-8K | N-16K | N-24K | N-32K | LongBench-v2-Short |
|---|---|---|---|---|---|---|
| Dream | 100.0 | 85.2 | 36.7 | 23.7 | 19.3 | 30.00 |
| Dream-NTK | 92.1 | 59.8 | 19.8 | 7.7 | 4.1 | 27.78 |
| Dream-Finetune | 100.0 | 100.0 | 100.0 | 100.0 | 100.0 | 31.11 |

**Key Findings.** Directly applying NTK scaling to the original Dream model without post-training substantially degrades long-context performance, even falling below the original 2K Dream baseline. In contrast, our training-based approach yields the strongest overall performance: Dream-Finetune consistently outperforms both the original Dream model and the directly scaled Dream-NTK variant on both NIAH-32K and LongBench-v2-Short. These results suggest that our training-based long-context scaling strategy generalizes effectively to diffusion LLMs derived from autoregressive models and remains robust across different training objectives.

## 5 CONCLUSION

In this paper, we discuss a lightweight, post-training route to scale the context window of diffusion LLM to up to 128K tokens. Our approach introduces a Diffusion-aware NTK extrapolation that accounts for bidirectional attention, and complements it with post-training strategies with advanced long-context data masking. Together, these components yield a long-context diffusion model, ULTRALLADA, where the experimental results show that ULTRALLADA substantially outperforms training-free scaling for diffusion LLMs and that both the diffusion-aware positional treatment and boundary-aware packing are necessary to unlock long-context competence.

## ACKNOWLEDGMENT

This work is supported by the HKUST startup grant R9895 from CSE; RGC-ECS project 26218024; RGC-NSFC project CRS_HKUST601/24. We thank the support from National Supercomputer Center in Guangzhou Nansha Sub-Center.

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

## A   THE USE OF LLMs IN WRITING

We used LLM, namely OPENAI-GPT5, to polish the writing of this manuscript. No other generative AI functionality is used in the writing of this submission.

## B   TRAINING SETTING DETAILS.

For reproducibility, we symmarize the training details and model hyperparameters in Table 7.

Table 7: Model training settings for all main results.

| | |
|---|---|
| **Model** | Initialization: LLaDA-8B-Base; 
 Tokenizer: the tokenizer used by LLaDA-8B-Base; 
 RoPE/NTK: `Diffuson-aware NTK` or `Baseline NTK` (as specified per experiment); 
 Attention: FlashAttention2 (Dao, 2023); full attention with `Adaptive Masking` (bias-based) when speicfied in experiment. |
| **Optim.** | AdamW (weight decay = 0.1, $\beta_1 = 0.9$, $\beta_2 = 0.95$); 
 LR: cosine, decay iters = 400, warmup = 3%, Grad clip = 1.0, peak LR = $2e^{-5}$, min LR = $2e^{-6}$; 
 Batch size: 4M tokens per iteration with 600 iterations; 
 Dropout: attention = 0.0, hidden = 0.0. |
| **Parallelism** | Megatron with data parallelism (Zero-2) and context parallelism; 
 trained on a 128-GPU cluster. |
| **Data** | PG19 (Rae et al., 2019) multi-document concatenation (64K per chunk); 
 `Direct Concatenation` or `EOD Caoncatenation` (as specified per experiment). |

## C   THEORETICAL ANALYSIS OF DIFFU-AWARE NTK.

Below, we provide a clearer theoretical justification, based on the structure of RoPE, for why a diffusion model trained on sequences of length $T_{\text{train}}$ naturally learns relative positions within the range $[-(T_{\text{train}} - 1), T_{\text{train}} - 1]$.

(1) RoPE makes attention scores depend on relative offsets $(j - i)$. Let the attention head dimension be $d$. We partition the query and key vectors into $d/2$ two-dimensional blocks:

$$\bar{\mathbf{q}} = (q_0, q_1, q_2, q_3, \ldots, q_{d-2}, q_{d-1})^\top, \qquad \bar{\mathbf{k}} = (k_0, k_1, k_2, k_3, \ldots, k_{d-2}, k_{d-1})^\top.$$

The $m$-th 2D block is associated with angular frequency $\theta_m$.

The RoPE transformation at position $i$ is defined as the block-diagonal matrix

$$R(i) = \text{diag}\big(R_0(i\theta_0), R_1(i\theta_1), \ldots, R_{d/2-1}(i\theta_{d/2-1})\big),$$

where each $2 \times 2$ rotation block is

$$R_m(i\theta_m) = \begin{pmatrix} \cos(i\theta_m) & -\sin(i\theta_m) \\ \sin(i\theta_m) & \cos(i\theta_m) \end{pmatrix}.$$

After applying positional encoding, the query at position $i$ and key at position $j$ become

$$\mathbf{q}_i = R(i)\bar{\mathbf{q}}, \qquad \mathbf{k}_j = R(j)\bar{\mathbf{k}}.$$

Their attention score is

$$\text{Attn}_{ij} = \mathbf{q}_i^\top \mathbf{k}_j = \bar{\mathbf{q}}^\top R(i)^\top R(j)\bar{\mathbf{k}}.$$

Using the group property of 2D rotations, for each block we have

$$R_m(i\theta_m)^\top R_m(j\theta_m) = R_m\big((j - i)\theta_m\big).$$

Therefore, the contribution of the $m$-th 2D block can be written as

$$\text{Attn}_{ij}^{(m)} = (\bar{\mathbf{q}}^{(m)})^\top R_m\big((j-i)\theta_m\big)\bar{\mathbf{k}}^{(m)} = A_m \cos\big((j-i)\theta_m\big) + B_m \sin\big((j-i)\theta_m\big),$$

where the content-dependent coefficients are

$$A_m = q_{2m}k_{2m} + q_{2m+1}k_{2m+1}, \qquad B_m = q_{2m}k_{2m+1} - q_{2m+1}k_{2m}.$$

Notably, $A_m$ and $B_m$ depend only on the content vectors and are independent of positions $i$ and $j$.

Summing over all 2D blocks, the full attention score becomes

$$\text{Attn}_{ij} = \sum_{m=0}^{d/2-1} \big[A_m \cos\big((j-i)\theta_m\big) + B_m \sin\big((j-i)\theta_m\big)\big].$$

This decomposition shows that all positional effects enter the attention score only through the relative offset $(j-i)$. In other words, RoPE explicitly parameterizes attention as a function of relative position.

(2) What range of relative positions does a diffusion model observe during training? During pre-training of a diffusion LLM, attention is bidirectional, so both query and key positions can range over the full training sequence:

$$i, j \in \{0, 1, \ldots, T_{\text{train}} - 1\}.$$

Therefore, the relative offset satisfies

$$j - i \in \big[-(T_{\text{train}} - 1),\, T_{\text{train}} - 1\big].$$

Since the RoPE attention score is explicitly a function of $(j-i)$, the diffusion model is directly exposed during training to relative positions over the entire range

$$\big[-(T_{\text{train}} - 1),\, T_{\text{train}} - 1\big].$$

Equivalently, the model learns a RoPE-induced mapping from relative offsets to attention scores over this full bidirectional domain.

## D  EMPIRICAL VALIDATION OF THE CRITICAL-DIMENSION ESTIMATE.

To further validate the analytical estimate of the diffusion-aware critical dimension, we conduct a finer-grained sweep over nearby settings and evaluate them on LongBench-v2-Short (tasks with context length $\leq$ 32K) and NIAH (8K–32K). The results are shown in Table 8. We observe that $d'_{\text{crit}} = 70$, as predicted by our analysis, is already near-optimal, while $d'_{\text{crit}} = 73$ achieves the best overall performance across the two benchmarks. This suggests that our analytical estimate provides a strong initialization point, and that the remaining discrepancy likely arises because different relative offsets are observed with non-uniform frequency during training.

Table 8: Finer-grained sweep over nearby critical dimensions.

| Critical Dimension | LongBench-v2-Short | NIAH (8K–32K) |
| --- | --- | --- |
| 69 | 27.22 | 60.97 |
| 70 | 28.89 | 69.26 |
| 71 | 28.89 | 71.01 |
| 73 | 31.11 | 73.23 |
| 76 | 27.78 | 72.84 |
| 79 | 27.78 | 63.39 |

**Evaluation Settings.** For RULER, we set the output length, block length, and number of denoising steps to 32. For LongBench-v2, we set the output length, block length, and number of denoising steps to 4. The prompt template used for LongBench-v2 is:

```
Please read the following text and answer the question below.

<text>
{context}
</text>

What is the correct answer to this question: {question}
Choices:
(A) {choice_A}
(B) {choice_B}
(C) {choice_C}
(D) {choice_D}

Answer:
```

# E  EFFECTIVENESS OF DIFFUSION-AWARE NTK

To further evaluate the effectiveness of diffusion-aware NTK under the same post-training setting, we extend the comparison between diffusion-aware NTK and baseline NTK to longer contexts beyond those reported in the main paper. Specifically, we evaluate:**RULER** with a context length of 48K; **LongBench-v2** with a context length of 64K.

As shown in Table 9, diffusion-aware NTK consistently outperforms baseline NTK across all evaluated tasks. Here, "LB-" denotes the corresponding LongBench-v2 task categories.

Table 9: Comparison between diffusion-aware NTK and baseline NTK on long-context evaluation.

| Method | LB-Easy | LB-Hard | LB-Short | LB-Medium | LB-Long | RULER |
|---|---|---|---|---|---|---|
| Diffusion-aware NTK | 35.94 | 28.94 | 37.22 | 27.91 | 29.63 | 56.94 |
| Baseline NTK | 33.33 | 27.97 | 35.56 | 27.44 | 25.93 | 53.75 |

These results indicate that incorporating diffusion-specific relative position spans yields more robust long-context behavior than directly applying AR-oriented NTK scaling.

**Evaluation Settings.** For RULER, we set the output length, block length, and number of denoising steps to 64. For LongBench-v2, we set the output length, block length, and number of denoising steps to 4. The prompt template used for LongBench-v2 is the same as that used in Appendix D.

# F  POST-TRAINING ON SHORT DATA.

We post-trained two variant models on short data (4K): one trained for 600 steps, aligned with long-context post-training settings, and another trained for 6000 steps to match the total number of tokens used in long-context training. Both models employed the Baseline NTK with the EOD concatenation strategy. We then evaluated them on RULER-4K, with results summarized in Table 10. Interestingly, while long-context post-training leads to higher scores compared to the base model, short-context post-training shows no improvement on RULER-4K. This may indicate that the performance gains of the extended-context model do not stem from additional training data, but may from the context-length extension itself.

Table 10: Post-training using short data.

| Model | AVG | NIAH | AGG | QA | VT |
|---|---|---|---|---|---|
| LLaDA-8B-Base | 86.17 | 99.19 | 45.84 | 67.5 | 100 |
| UltraLLaDA | 88.37 | 97.31 | 66.69 | 68.50 | 100 |
| 4K-600steps | 85.74 | 99.31 | 48.49 | 62.5 | 98.2 |
| 4K-6000steps | 84.16 | 98.5 | 51.37 | 53.50 | 96.4 |

## G   EXTRA EXPERIMENTS ON LONGBENCH-4K.

At the 4K context length, all settings deliver comparable performance. Notably, a similar pattern to that reported in Appendix F, where context-length scaling results in higher scores than base model.

Table 11: Extra LongBench-4K experiment results.

| Model | AVG | SD | MD | Sum | ICL | Syn | Code |
|---|---|---|---|---|---|---|---|
| LLaDA-8B-Base | 33.45 | 14.35 | 12.58 | 25.62 | 65.08 | 28.32 | 59.71 |
| UltraLLaDA | 35.55 | 16.67 | 12.73 | 25.78 | 65.38 | 29.9 | 67.25 |
| Baseline-NTK + EOD-cat | 34.93 | 16.12 | 12.32 | 24.95 | 64.42 | 28.98 | 67.03 |
| Diffu-NTK + EOD-cat | 35.43 | 15.73 | 12.81 | 26.29 | 65.60 | 29.00 | 67.22 |
| Diffu-NTK + Adapt-Mask | 35.55 | 16.67 | 12.73 | 25.78 | 65.38 | 29.9 | 67.25 |
| Diffu-NTK + Direct-Cat | 35.23 | 15.54 | 12.67 | 26.43 | 64.64 | 29.63 | 66.48 |

## H   ADDITIONAL LONG-CONTEXT EVALUATIONS BEYOND PPL AND RETRIEVAL

To further assess the robustness of ULTRALLADA beyond perplexity and simple retrieval-based evaluations, we expand our evaluation to more complex long-context comprehension and reasoning tasks at longer context lengths. Specifically, we conduct additional experiments on RULER with a context length of 48K, and on LongBench-v2 with context lengths of 64K and 96K.

Table 12 summarizes the performance of ULTRALLADA on RULER-48K, and Table 13 reports results on LongBench-v2 at 64K and 96K.

Table 12: Performance of ULTRALLADA on RULER with a 48K context length.

| UltraLLaDA | AGG | QA | VT | NIAH |
|---|---|---|---|---|
| RULER-48K | 25.12 | 24.50 | 95.20 | 68.22 |

Table 13: Performance of ULTRALLADA on LongBench-v2 at longer context lengths.

| UltraLLaDA | Easy | Hard | Short | Medium | Long |
|---|---|---|---|---|---|
| LongBench-v2-64K | 35.94 | 28.94 | 37.22 | 27.91 | 29.63 |
| LongBench-v2-96K | 38.54 | 27.97 | 37.22 | 27.77 | 27.78 |

These benchmarks cover a broader range of long-context comprehension and reasoning tasks than perplexity and simple needle-in-a-haystack retrieval. The results show that ULTRALLADA remains robust as the context length increases, indicating that its gains extend beyond retrieval-heavy settings and generalize to more diverse long-context reasoning scenarios.

**Evaluation Settings.**   The evaluation configuration and prompt template are the same as those used in the Appendix D.

## I   TRADITIONAL SHORT CONTEXT BENCHMARK CASE STUDY.

We evaluated both the base model and the context-extended model on Winogrande (Sakaguchi et al., 2021) and ARC-c (Clark et al., 2018). For Winogrande, we used a 5-shot setting with two generated tokens and a sampling step of 2. For ARC-c, we adopted a zero-shot setting with the same generation configuration. The results, summarized in Table 14, show that the context-extended model does not outperform the base model on either benchmark. This may be because, similar to auto-regressive models, Diffusion LLMs also tend to experience performance degradation on tasks involving more complex reasoning after context-length extension.

Table 14: Other Benchmarks.

| Model | AVG | Winogrande | ARC-c |
|---|---|---|---|
| LLaDA-8B-Base | 79.95 | 78.53 | 81.36 |
| UltraLLaDA | 77.42 | 73.48 | 81.36 |
| Baseline-NTK + EOD-cat | 76.10 | 72.53 | 79.66 |
| Diffu-NTK + EOD-cat | 76.35 | 72.69 | 80.00 |

## J DETAILS ON TRAINING-FREE CASE STUDY.

We provide detailed experiment results on training-free case study in Section 3.2.

**NIAH.** We compare Diffusion-aware NTK against the Baseline NTK on the NIAH retrieval task across context lengths from 4K to 32K tokens. The Diffusion-aware NTK obtains a retrieval accuracy of 79.36%, outperforming the Baseline NTK (75.68%).

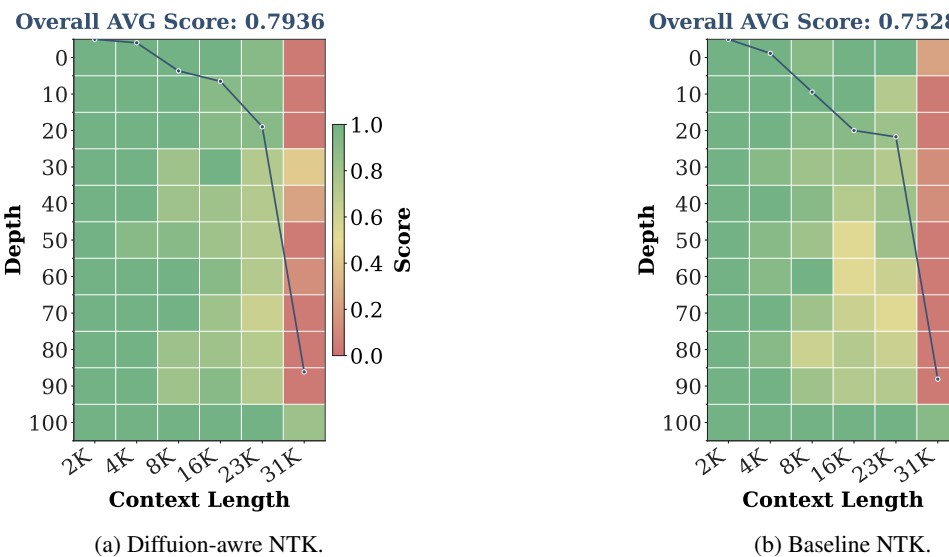

(a) Diffuion-awre NTK.  (b) Baseline NTK.

Figure 4: NIAH evaluation from 4K to 32K context-length.

**LongBench.** We compare Diffusion-aware NTK with Baseline NTK on the LongBench (cut at 16K) benchmark. Diffusion-aware NTK slightly outperfomrs the baseline.

Table 15: LongBench16K training-free case study.

| Model | AVG | SD | MD | Sum | ICL | Syn | Code |
|---|---|---|---|---|---|---|---|
| Baseline NTK | 35.02 | 14.91 | 11.71 | 24.92 | 66.30 | 39.62 | 60.55 |
| Diffu-aware NTK | 35.38 | 15.14 | 11.26 | 24.79 | 67.22 | 40.56 | 61.53 |

