# OpenReview forum: "UltraLLaDA: Scaling the Context Length to 128K for Diffusion Large Language Models"
_ICLR.cc/2026/Conference — ICLR 2026 Poster_

### Official Review · Reviewer_S1Du · 2025-10-27

**Soundness:** 2
**Presentation:** 3
**Contribution:** 3
**Rating:** 6
**Confidence:** 5

**Summary:**

This paper introduces UltraLLaDA, a method for extending the context window of diffusion-based Large Language Models (LLMs) to 128,000 tokens through an efficient post-training process. The research addresses a critical and largely unexplored area, as the long-context capabilities of diffusion LLMs have not been systematically studied.

The authors propose two main contributions:

Diffusion-aware NTK: A novel adaptation of the Neural Tangent Kernel (NTK) method for scaling Rotary Positional Embeddings (RoPE). The key insight is that diffusion models, with their bidirectional attention mechanism, learn a much wider range of relative positions during pre-training (approximately twice the context length) compared to auto-regressive models. By accounting for this property, the authors develop a more suitable RoPE scaling factor that enables stable extrapolation to very long contexts.

Masking Strategy Analysis: The paper systematically investigates data packing and attention masking strategies to mitigate "cross-document interference" during long-context fine-tuning—a significant challenge for models with global bidirectional attention. It compares adaptive attention masking (which blocks attention between concatenated documents) and end-of-document (EOD) token concatenation against a naive direct concatenation baseline.

Empirically, UltraLLaDA demonstrates remarkable performance. It achieves 100% accuracy on the "Needle-in-a-Haystack" (NIAH) retrieval task at the full 128K context length. Across a suite of benchmarks, including Perplexity, LongBench, and RULER, UltraLLaDA consistently and significantly outperforms the base LLaDA model and a training-free extension baseline (LongLLaDA), with the performance gap widening as context length increases. Ablation studies confirm that both the Diffusion-aware NTK and the use of boundary-aware masking strategies are essential for achieving these results.

**Strengths:**

Addresses a novel and important problem: long-context extension for diffusion LLMs. As diffusion models gain traction, understanding how to scale their context window is crucial for their practical application and competitiveness.

The model's performance is a standout feature. Achieving perfect 100% accuracy on the 128K NIAH task is a powerful demonstration of the method's effectiveness in long-range information retrieval. The consistent and significant improvements over baselines across multiple diverse benchmarks (PPL, LongBench, RULER) provide robust evidence supporting the authors' claims.

The core technical contributions are simple, intuitive, and clearly justified. The adaptation of NTK scaling is based on a clear-eyed observation of the architectural differences between diffusion and auto-regressive models. Furthermore, the systematic study of masking strategies provides valuable, practical insights for training such models.

The paper includes thorough ablation studies that successfully isolate the impact of each key component (the NTK variant and the masking strategy). This experimental rigor greatly strengthens the validity of the conclusions and clearly demonstrates that both proposed techniques are necessary for the final performance.

**Weaknesses:**

The narrow set of baselines, all comparisons are internal to the diffusion model LLaDA and LongLLaDA. Considering that this is relatively new in diffusion LLM, this is understandable, but perhaps could consider migrating other methods commonly used in auto-regression models for comparison, such as PI and YARN, to provide more insights.

The appendix reveals that UltraLLaDA's performance on standard short-context benchmarks degrades after long-context fine-tuning. This is a critical trade-off common in context extension methods but is not addressed in the main body of the paper. Acknowledging and analyzing this limitation in the main text would provide a more balanced and complete picture of the method's characteristics.

The evaluation could be strengthened by incorporating more challenging long-context reasoning benchmarks to provide a more comprehensive assessment of the model's capabilities beyond information retrieval.

**Questions:**

Q1: In Section 3.2, the explanation provided for $T_{\text{cap}}$ and $T_{\text{Ecap}}$ being twice as large in diffusion LLMs compared to auto-regressive LLMs is intuitive. However, the argument would be significantly strengthened by a more detailed theoretical derivation or formal proof to rigorously support this conclusion.

Q2: The paper compares the model's training-free performance at critical dim = 64 versus 70. It would be insightful to see an analysis of performance at other proximal values (e.g., 69 or 71). Investigating whether 70 represents an optimal or near-optimal setting would provide stronger validation for the optimization process of the Diffusion-aware NTK method.

Q3: The evaluation of long-context capabilities currently concentrates primarily on retrieval tasks. To provide a more comprehensive assessment, please consider including benchmarks that evaluate long-context reasoning abilities, such as the relevant tasks in LongBench v2.

Q4: The experiments cap the maximum sequence length extension at 128k. Could the authors clarify the rationale for this specific limit? It would be beneficial to understand the method's performance at even greater lengths or, alternatively, to discuss the method's ultimate scaling limit (e.g., what is the maximum extension factor this approach can effectively achieve?).

---

> ### Author Response · Authors · 2025-11-21
>
> > Q1: In Section 3.2, the explanation provided for  and  being twice as large in diffusion LLMs compared to auto-regressive LLMs is intuitive. However, the argument would be significantly strengthened by a more detailed theoretical derivation or formal proof to rigorously support this conclusion.
> >
>
> We thank the reviewer for the suggestion. Below, we provide a clearer theoretical justification, based on the structure of RoPE, for why a diffusion model trained on sequences of length $T_{train}$ naturally learns relative positions within the range $[-(T_{train}-1),T_{train}-1].$
>
> **(1) RoPE makes attention scores depend on relative offsets $(j-i).$**
>
> Let the head dimension be $d$. We split the query and key into $d/2$ two-dimensional groups:
>
> $$
> \bar{q} = (q_{0},q_{1},q_{2},q_{3},...,q_{d-2},q_{d-1})^{T},\\
> \bar{k} = (k_{0},k_{1},k_{2},k_{3},...,k_{d-2},k_{d-1})^{T},\\
> $$
>
> The $m$-th 2D group is rotated with frequency $\theta_{m}$.
>
> The RoPE rotation for position $i$ is a block-diagonal matrix:
>
> $$
> R(i) = diag(R_{0}(i\theta_{0}),R_{1}(i\theta_{1}),...,R_{d/2-1}(i\theta_{d/2-1})),\\
> R_{m}(i\theta_{m}) = \begin{pmatrix} \cos(i\theta_{m}) & -\sin(i\theta_m) \\ \sin(i\theta_m) & \cos(i\theta_m)\end{pmatrix}.
> $$
>
> After position encoding:
>
> $$
> q_{i} = R(i)\bar{q}, \quad k_{j}=R(j)\bar{k}.
> $$
>
> The attention score between the $i$-th query and $j$-th key becones
>
> $$
> Attn_{ij}=q_{i}^{T}k_{j}=\bar{q}^{T}R(i))^{T}R(j)\bar{k}.
> $$
>
> For each 2D block,
>
> $$
> Attn_{ij}^{(m)}=(\bar{q}^{(m)})^{T}R_m((j-i)\theta_{m})\bar{k}^{(m)} = A_{m}\cos((j-i)\theta_{m})+B_{m}\sin((j-i)\theta_{m}),
> $$
>
> where the content-dependent coefficients
>
> $$
> A_m = q_{2m}k_{2m} + q_{2m+1}k_{2m+1},\quad B_{m} = q_{2m}k_{2m+1}-q_{2m+1}k_{2m}
> $$
>
> do not depend on positions.
>
> Thus, full attention is
>
> $$
> Attn_{ij} = \sum_{m=0}^{d/2-1}(A_{m}\cos((j-i)\theta_{m})+B_{m}\sin((j-i)\theta_{m})).
> $$
>
> This decomposition shows:
>
> - All positional effects enter **only** through $(j-i).$.
> - RoPE explicitly parameterizes attention as a function of the **relative offset**.
>
> **(2) What range of relative positions does the diffusion model see during training?**
>
> In pre-training of diffusion LLMs, all attention pairs satisfy
>
> $$
> i,j\in \set{0,...,T_{train}-1}.
> $$
>
> Therefore, the model is exposed to relative offsets:
>
> $$
> (j-i)\in [-(T_{train}-1), T_{train}-1].
> $$
>
> Because attention is a sum of terms over $(j-i)$, the diffusion model RoPE learns a function: $f_{RoPE}:(j-i) \mapsto Attn_{ij}$ directly over this offset domain.

---

> ### Author Response · Authors · 2025-11-21
>
> > Q2: The paper compares the model's training-free performance at critical dim = 64 versus 70. It would be insightful to see an analysis of performance at other proximal values (e.g., 69 or 71). Investigating whether 70 represents an optimal or near-optimal setting would provide stronger validation for the optimization process of the Diffusion-aware NTK method.
> >
>
> We appreciate the reviewer’s suggestion. Following this advice, we conducted a finer-grained sweep of nearby settings and evaluated them on **LongBench-v2-short (≤32K)** and **NIAH (8K~32K)**.
>
> The results are summarized below:
>
> |  | longbench-v2-short(≤32K) | NIAH-(8K~32K) |
> | --- | --- | --- |
> | critical dimension 69 | 27.22 | 60.97 |
> | critical dimension 70 | 28.89 | 69.26 |
> | critical dimension 71 | 28.89 | 71.01 |
> | critical dimension 73 | 31.11 | 73.23 |
> | critical dimension 76 | 27.78 | 72.84 |
> | critical dimension 79 | 27.78 | 63.39 |
>
> We observe that **70** is indeed near-optimal, while **73** performs best jointly across the two benchmarks. These results suggest that our analytical estimate provides a good starting point and that remaining deviations likely come from the non-uniform frequency with which different relative offsets are observed during training.
>
> **Evaluation Settings.**
>
> - NIAH. We set output length, block length, and denoising step all to 32;
> - LongBench-v2. We set output length, block length, and denoising step all to 4. The prompt template is: 'Please read the following text and answer the question below.\n\n<text>\n{context}\n</text>\n\nWhat is the correct answer to this question: {question}\nChoices:\n(A) {choice_A}\n(B) {choice_B}\n(C) {choice_C}\n(D) {choice_D}\n\nAnswer:'

---

> ### Author Response · Authors · 2025-11-21
>
> > Q3: The evaluation of long-context capabilities currently concentrates primarily on retrieval tasks. To provide a more comprehensive assessment, please consider including benchmarks that evaluate long-context reasoning abilities, such as the relevant tasks in LongBench v2.
> >
>
> > W3: The evaluation could be strengthened by incorporating more challenging long-context reasoning benchmarks to provide a more comprehensive assessment of the model's capabilities beyond information retrieval.
> >
>
> We appreciate the reviewer’s suggestion. We expanded our evaluation to include more complex comprehension and reasoning tasks at longer context lengths.
>
> **(1) Additional long-context evaluations beyond PPL and simple NIAH.**
>
> We conducted new experiments on:
>
> - RULER with 48K context length.
> - LongBengch-v2 with 64K and 96K context length.
>
> UltraLLaDA’s performance is summarized below:
>
> | UltraLLaDA | AGG | QA | VT | NIAH |
> | --- | --- | --- | --- | --- |
> | ruler-48K | 25.12 | 24.5 | 95.2 | 68.22 |
>
> | UltraLLaDA | Easy | Hard | Short | Medium | Long |
> | --- | --- | --- | --- | --- | --- |
> | long bench-v2-64K | 35.94 | 28.94 | 37.22 | 27.91 | 29.63 |
> | longbench-v2-96K | 38.54 | 27.97 | 37.22 | 27.77 | 27.78 |
>
> These benchmarks contain diverse tasks requiring reasoning and understanding. The results show that UltraLLaDA maintains performance when scaling to longer contexts, demonstrating its robustness beyond information-retrieval and perplexity-based metrics.
>
> **Evaluation Settings.**
>
> - RULER. We set output length, block length, and denoising step all to 64;
> - LongBench-v2. evaluation configuration and prompt template are the same as the response to Q2.

---

> ### Author Response · Authors · 2025-11-21
>
> > Q4: The experiments cap the maximum sequence length extension at 128k. Could the authors clarify the rationale for this specific limit? It would be beneficial to understand the method's performance at even greater lengths or, alternatively, to discuss the method's ultimate scaling limit (e.g., what is the maximum extension factor this approach can effectively achieve?).
> >
>
> We thank the reviewer for the question.
>
> **(1) Why 128K?**
>
> Community convention. In the autoregressive literature, 128K is widely regarded as a standard “long-context” regime, and many representative long-context LLM works report their main results at ≤128K.
>
> Scaling diffusion LLMs beyond 128K introduces challenges that are substantially more severe than in AR models:
>
> - Inference efficiency bottleneck.
> Unlike AR models with KV cache, diffusion LLMs recompute attention over the entire sequence at every denoising step. Thus, the inference cost scales roughly as $O(T^2 \times \text{steps})$. At 512K–1M tokens, the cost becomes prohibitive for our hardware. We are now actively researching methods to mitigate this.
> - Resource constraints for training.
>
>     Training on long sequences would require vast GPU memory and compute; even with activation-checkpointing and other optimization strategies, the resource consumption grows beyond what our current infrastructure can support.
>
>
> **(2) Scaling limit.**
>
> Conceptually, our training-based method does not impose a hard upper bound on achievable context length. We therefore treat 128K as a practical long-context target that is (i) commonly adopted in prior AR research, and (ii) feasible to train and evaluate under the constraints of diffusion LLMs today.

---

> ### Author Response · Authors · 2025-11-21
>
> > W1: The narrow set of baselines, all comparisons are internal to the diffusion model LLaDA and LongLLaDA. Considering that this is relatively new in diffusion LLM, this is understandable, but perhaps could consider migrating other methods commonly used in auto-regression models for comparison, such as PI and YARN, to provide more insights.
> >
>
> We thank the reviewer’s suggestion. Our choice of baselines is mainly driven by the current state of long‑context scaling in diffusion LLMs and by the specific question we study.
>
> 1. Although many RoPE scaling strategies (PI, NTK, YaRN, etc.) have been extensively studied in autoregressive LLMs, RoPE scaling for diffusion LLMs remains almost entirely unexplored. To date, LongLLaDA is the only prior work, and it exclusively adopts NTK scaling. Thus, **NTK is currently the only method empirically validated to work in diffusion LLMs**, which makes it the most natural choice of baseline in our study. Building on LongLLaDA’s NTK formulation allows us to keep the comparison controlled when we move from a training‑free to a training‑based extension.
> 2. The goal of our paper is to address: **Can training-based context extension be effective for diffusion LLMs at all?** This question is non-trivial because the language modeling and learning objectives of AR and diffusion models differ fundamentally—AR LLMs rely on next-token prediction, while diffusion LLMs rely on iterative denoising conditioned on noised latent states. Whether the diffusion training objective can effectively support long-context scaling was unknown prior to our investigation. Therefore, we intentionally isolate this factor by adopting NTK—the only DLLM-tested scaling method—to ensure a controlled comparison between **training-free** and **training-based** strategies.
>
> Finally, our choice does not imply that PI, YaRN, or other fine-grained AR-developed scaling methods are incompatible with diffusion models. Rather, these approaches introduce depth-dependent or frequency-modulated behaviors whose interactions with the diffusion denoising process and training objectives remain unexplored. Adapting AR-designed scaling mechanisms to diffusion LLms is itself an oen research problem, and we consider it a promising direction fo future work beyond the scope of this study.

---

> ### Author Response · Authors · 2025-11-21
>
> > W2: The appendix reveals that UltraLLaDA's performance on standard short-context benchmarks degrades after long-context fine-tuning. This is a critical trade-off common in context extension methods but is not addressed in the main body of the paper. Acknowledging and analyzing this limitation in the main text would provide a more balanced and complete picture of the method's characteristics.
> >
>
> We thank the reviewer’s suggestion. Indeed, similar to autoregressive LLMs, extending the context window of diffusion LLMs can introduce a trade-off between long-context gains and short-context performance. Our appendix reports this degradation, and we agree that acknowledging it in the main paper would provide a more balanced perspective.
>
> In the revision, we will explicitly discuss this limitation and provide clarification:
>
> - The degradation is **expected and consistent** with prior findings in AR models, where large context extrapolation often shifts model capacity toward modeling long-range dependencies.
> - Our current work focuses on demonstrating the feasibility and effectiveness of training-based long-context scaling for diffusion LLMs.
> - Addressing the short-context regression—e.g., through joint optimization with short-context data—is a promising direction for future research.
>
> We appreciate the reviewer for pointing out this trade-off, and we will make it clearer in the main paper to give a more complete characterization of UltraLLaDA.

---

> ### Comment · Reviewer_S1Du · 2025-11-28
>
> I appreciate the authors' detailed response. Since the rebuttal has effectively addressed my questions, I will keep my original rating.

---

### Official Review · Reviewer_ev1L · 2025-10-30

**Soundness:** 3
**Presentation:** 3
**Contribution:** 2
**Rating:** 6
**Confidence:** 4

**Summary:**

This paper presents UltraLLaDA, a method for extending the context length of diffusion-based large language models to 128K tokens. The approach introduces a diffusion-aware NTK scaling technique and explores various masking strategies for handling long-context documents. Experimental results demonstrate the effectiveness of the proposed method.

**Strengths:**

1. The paper is well-written and easy to follow.

2. The proposed diffusion-aware NTK scaling method is simple yet effective, showing a certain level of novelty.

3. The paper conducts extensive experiments that clearly demonstrate the effectiveness of the proposed approach.

**Weaknesses:**

1. The proposed diffusion-aware NTK scaling is primarily based on an empirical assumption that a diffusion LLM can naturally handle a wider range of relative positions, with $T_{cap} \sim 2T_{train}$ and $T_{Ecap} \sim 2T_{target}$. It would be more convincing if the paper provided deeper theoretical justification or analysis for this assumption.

2. The investigation of different masking techniques for long documents shows limited novelty, as similar findings have already been discussed in prior studies on large language models[1,2].

[1] LongRoPE2: Near-Lossless LLM Context Window Scaling

[2]  The Llama 3 Herd of Models, https://arxiv.org/abs/2407.21783

**Questions:**

See the weaknesses section

---

> ### Author Response · Authors · 2025-11-21
>
> > W1: The proposed diffusion-aware NTK scaling is primarily based on an empirical assumption that a diffusion LLM can naturally handle a wider range of relative positions, with and . It would be more convincing if the paper provided deeper theoretical justification or analysis for this assumption.
> >
>
> We thank the reviewer for the suggestion. Below, we provide a clearer theoretical justification, based on the structure of RoPE, for why a diffusion model trained on sequences of length $T_{train}$ naturally learns relative positions within the range $[-(T_{train}-1),T_{train}-1].$
>
> **(1) RoPE makes attention scores depend on relative offsets $(j-i).$**
>
> Let the head dimension be $d$. We split the query and key into $d/2$ two-dimensional groups:
>
> $$
> \bar{q} = (q_{0},q_{1},q_{2},q_{3},...,q_{d-2},q_{d-1})^{T},\\
> \bar{k} = (k_{0},k_{1},k_{2},k_{3},...,k_{d-2},k_{d-1})^{T},\\
> $$
>
> The $m$-th 2D group is rotated with frequency $\theta_{m}$.
>
> The RoPE rotation for position $i$ is a block-diagonal matrix:
>
> $$
> R(i) = diag(R_{0}(i\theta_{0}),R_{1}(i\theta_{1}),...,R_{d/2-1}(i\theta_{d/2-1})),\\
> R_{m}(i\theta_{m}) = \begin{pmatrix} \cos(i\theta_{m}) & -\sin(i\theta_m) \\ \sin(i\theta_m) & \cos(i\theta_m)\end{pmatrix}.
> $$
>
> After position encoding:
>
> $$
> q_{i} = R(i)\bar{q}, \quad k_{j}=R(j)\bar{k}.
> $$
>
> The attention score between the $i$-th query and $j$-th key becones
>
> $$
> Attn_{ij}=q_{i}^{T}k_{j}=\bar{q}^{T}R(i))^{T}R(j)\bar{k}.
> $$
>
> For each 2D block,
>
> $$
> Attn_{ij}^{(m)}=(\bar{q}^{(m)})^{T}R_m((j-i)\theta_{m})\bar{k}^{(m)} = A_{m}\cos((j-i)\theta_{m})+B_{m}\sin((j-i)\theta_{m}),
> $$
>
> where the content-dependent coefficients
>
> $$
> A_m = q_{2m}k_{2m} + q_{2m+1}k_{2m+1},\quad B_{m} = q_{2m}k_{2m+1}-q_{2m+1}k_{2m}
> $$
>
> do not depend on positions.
>
> Thus, full attention is
>
> $$
> Attn_{ij} = \sum_{m=0}^{d/2-1}(A_{m}\cos((j-i)\theta_{m})+B_{m}\sin((j-i)\theta_{m})).
> $$
>
> This decomposition shows:
>
> - All positional effects enter **only** through $(j-i).$.
> - RoPE explicitly parameterizes attention as a function of the **relative offset**.
>
> **(2) What range of relative positions does the diffusion model see during training?**
>
> In pre-training of diffusion LLMs, all attention pairs satisfy
>
> $$
> i,j\in \set{0,...,T_{train}-1}.
> $$
>
> Therefore, the model is exposed to relative offsets:
>
> $$
> (j-i)\in [-(T_{train}-1), T_{train}-1].
> $$
>
> Because attention is a sum of terms over $(j-i)$, the diffusion model RoPE learns a function: $f_{RoPE}:(j-i) \mapsto Attn_{ij}$ directly over this offset domain.

---

> ### Author Response · Authors · 2025-11-21
>
> > W2: The investigation of different masking techniques for long documents shows limited novelty, as similar findings have already been discussed in prior studies on large language models[1,2].
> >
>
> We thank the reviewer for pointing out the concern. While similar masking or packing strategies have indeed been explored in autoregressive LLMs, our contribution lies in systematically evaluating these techniques, for the first time, in the context of diffusion LLMs.
>
> In our experiments, we observe interesting behaviors in the DLLM training paradigm:
>
> - **EOD Concatenation** performs better at short to moderate lengths (8K~16K). At these lengths, the model can still attend globally, and the lighter constraint introduces beneficial flexibility.
> - **Adaptive Masking**, in contrast, becomes crucial at very long lengths (e.g., ≥32K). For DLLMs, unrestricted cross-document attention at long ranges leads to semantic interference. Adaptive Masking fully removes this by preventing cross-document interactions.
>
> Based on our empirical observations, we provide practical guidance: Since long-context scaling for DLLMs typically targets lengths beyond 32K, **Adaptive Masking is the more reliable strategy**, offering consistent protection against cross-document interference. Both Adaptive Masking and EOD Concatenation outperform Direct Concatenation, underscoring the importance of explicitly modeling document boundaries when retraining diffusion LLMs for long contexts.

---

### Official Review · Reviewer_eBLu · 2025-11-01

**Soundness:** 3
**Presentation:** 3
**Contribution:** 2
**Rating:** 6
**Confidence:** 4

**Summary:**

This paper introduces UltraLLaDA, a post-training approach designed to extend the context window of diffusion-based large language models (specifically LLaDA) from 4K to 128K tokens. The authors make two main contributions: first, they develop a diffusion-aware NTK scaling method that refines standard RoPE extrapolation to better suit the bidirectional attention mechanisms of diffusion models, using a context cap of approximately twice the training length rather than the single-length limit typical of autoregressive models. Second, they explore several masking strategies for managing multi-document concatenation during training, comparing adaptive masking, the use of explicit end-of-document (EOD) tokens, and direct concatenation. UltraLLaDAdemonstrates strong performance on long-context benchmarks such as NIAH, LongBench, and RULER, outperforming the training-free baseline, LongLLaDA.

**Strengths:**

1. Systematic study of post-training methods for extending context in diffusion LLMs, addressing an underexplored area.
2. Well-motivated technical approach. The diffusion-aware NTK modification is intuitive, accounting for bidirectional attention means the model sees roughly 2x the relative position range during training.
3. Comprehensive evaluation. Multiple benchmarks (PPL, NIAH, LongBench, RULER) across context lengths up to 128k tokens
4. Lightweight post-training (600 steps) makes the approach accessible.

**Weaknesses:**

1. Single base model. All experiments use only LLaDA-8b as the base. Generalization to other diffusion LLMs or different model sizes is unclear.
2. LongLLaDA baseline cannot be evaluated beyond 32k, making comparisons incomplete at the longest contexts.
3. Masking strategy conclusions unclear. Tables 4-5 show adaptive masking and EOD concatenation trading advantages at different lengths, but the paper doesn't provide clear guidance on which to use when. The difference between all methods are very small and could be attributed to noise.
4. No comparison against YaRN, a more widely used RoPE interpolation method.

**Questions:**

1. Can you provide theoretical or empirical analysis showing that diffusion models actually learn relative positions in the range [-2T_train, 2T_train] during pre-training?
2. Why was NTK-aware/ABF RoPE scaling used instead YaRN? Is there a intrinsic problem with YaRN that prevents it from being adapted to diffusion models? SoTA models are mostly always using YaRN instead of NTK scaling.
3. Have you tested this approach on other diffusion LLMs or other model sizes? Would it work without finetuning on larger model sizes? What about other architectures?

---

> ### Author Response · Authors · 2025-11-21
>
> > W1: Single base model. All experiments use only LLaDA-8b as the base. Generalization to other diffusion LLMs or different model sizes is unclear.
> >
>
> > Q3: Have you tested this approach on other diffusion LLMs or other model sizes? Would it work without finetuning on larger model sizes? What about other architectures?
> >
>
> We thank the reviewer for raising the concern. To address this, we extended our study beyond LLaDA and applied our training-based scaling approach to the **Dream model** (Dream-v0-Base-7B), a representative diffusion LLM converted from an autoregressive model.
>
> **(1) Applying our training-based method to Dream.**
>
> **Setup.**
>
> Dream has a native **2K** context and retains an AR-style shifted-label architecture, which is structurally different from LLaDA’s unshifted diffusion training. We extend Dream from 2K to 32K using our training-based pipeline and compare:
>
> 1. **Dream**: original 2K model.
> 2. **Dream-NTK**: direct NTK scaling (LongLLaDA-style) without any training.
> 3. **Dream-finetune (32K Dream)**: Dream post-trained on 32K data with our method.
>
> We evaluate on **NIAH-32K** and **LongBench-v2-short (≤32K)**:
>
> |  | N-4K | N-8K | N-16K | N-24K | N-32K | long bench-v2-short(≤32K) |
> | --- | --- | --- | --- | --- | --- | --- |
> | Dream | 100 | 85.2 | 36.7 | 23.7 | 19.3 | 30.00 |
> | Dream-ntk | 92.1 | 59.8 | 19.8 | 7.7 | 4.1 | 27.78 |
> | Dream-finetune | 100 | 100 | 100 | 100 | 100 | 31.11 |
>
> （The evaluation configuration and prompt template are the same as the response to W1.）
>
> **(2) Key empirical finding.**
>
> - Directly applying **NTK-scale (LongLLaDA’s method)** to the original, untrained Dream model **degrades** long-context performance—below even the original 2K Dream baseline.
> - In contrast, with our **training-based approach**, the 32K Dream model achieves **the best overall performance** among the three variants: Original Dream, NTK-scaled Dream, and our 32K-Dream.
>
> This demonstrates that the **training-based strategy generalizes successfully** to diffusion LLMs that originate from AR models and use different training objectives.
>
> **(3) On model sizes.**
>
> We acknowledge the reviewer’s concern regarding the larger model size. However, at present, **8B** is the largest open-source diffusion LLM publicly available.

---

> ### Author Response · Authors · 2025-11-21
>
> > W2: LongLLaDA baseline cannot be evaluated beyond 32k, making comparisons incomplete at the longest contexts.
> >
>
> We thank the reviewer for pointing out the concern. To address it, we expanded our evaluation to include **more complex comprehension and reasoning tasks** at longer context lengths.
>
> **(1) Additional long-context evaluations beyond PPL and simple NIAH.**
>
> We conducted new experiments on:
>
> - RULER with 48K context length.
> - LongBengch-v2 with 64K and 96K context length.
>
> UltraLLaDA’s performance is summarized below:
>
> | UltraLLaDA | AGG | QA | VT | NIAH |
> | --- | --- | --- | --- | --- |
> | ruler-48K | 25.12 | 24.5 | 95.2 | 68.22 |
>
> | UltraLLaDA | Easy | Hard | Short | Medium | Long |
> | --- | --- | --- | --- | --- | --- |
> | long bench-v2-64K | 35.94 | 28.94 | 37.22 | 27.91 | 29.63 |
> | longbench-v2-96K | 38.54 | 27.97 | 37.22 | 27.77 | 27.78 |
>
> The evaluation configuration and prompt template are the same as the response to W1.
>
> These benchmarks contain diverse tasks requiring reasoning and understanding. The results show that UltraLLaDA maintains performance when scaling to longer contexts, demonstrating its robustness beyond information-retrieval and perplexity-based metrics.

---

> ### Author Response · Authors · 2025-11-21
>
> > W3: Masking strategy conclusions unclear. Tables 4-5 show adaptive masking and EOD concatenation trading advantages at different lengths, but the paper doesn't provide clear guidance on which to use when. The difference between all methods are very small and could be attributed to noise.
> >
>
> We thank the reviewer for raising this question. To clarify the conclusion, we provide a more structured interpretation and practical guidance:
>
> **(1) Why do the two strategies behave differently?**
>
> - **EOD Concatenation** performs slightly better at short to moderate lengths (e.g., 8K - 16K). At these lengths, the model can still attend globally, and simply making document boundaries is sufficient. The lighter constraint also provides the model with more flexibility, which seems beneficial when the context lengths are not extremely long.
> - **Adaptive Masking** performs best at very long lengths (e.g., 32K). When the sequence becomes very long, unrestricted cross-document attention can introduce semantic interference across unrelated documents. Adaptive Masking fully avoids this issue by blocking cross-document attention, which becomes important when the sequence length is long.
>
> **(2) Practical guidance.**
>
> To make the conclusion explicit:
>
> Considering that scaling diffusion LLMs typically targets context lengths beyond 32K, **Adaptive Masking is generally the preferred strategy**, as it provides the most reliable protection against cross-document interference at these longer scales. Notably, both Adaptive Masking and EOD Concatenation outperform Direct Concatenation, underscoring the importance of explicitly modeling document boundaries during long-context training.

---

> ### Author Response · Authors · 2025-11-21
>
> > W4: No comparison against YaRN, a more widely used RoPE interpolation method.
> >
>
> > Q2: Why was NTK-aware/ABF RoPE scaling used instead YaRN? Is there a intrinsic problem with YaRN that prevents it from being adapted to diffusion models? SoTA models are mostly always using YaRN instead of NTK scaling.
> >
>
> We thank the reviewer for raising the question. Our choice of baselines is mainly driven by the current state of long‑context scaling in diffusion LLMs and by the specific question we study.
>
> 1. Although many RoPE scaling strategies (PI, NTK, YaRN, etc.) have been extensively studied in autoregressive LLMs, RoPE scaling for diffusion LLMs remains almost entirely unexplored. To date, LongLLaDA is the only prior work, and it exclusively adopts NTK scaling. Thus, **NTK is currently the only method empirically validated to work in diffusion LLMs**, which makes it the most natural choice of baseline in our study. Building on LongLLaDA’s NTK formulation allows us to keep the comparison controlled when we move from a training‑free to a training‑based extension.
> 2. The goal of our paper is to address: **Can training-based context extension be effective for diffusion LLMs at all?** This question is non-trivial because the language modeling and learning objectives of AR and diffusion models differ fundamentally—AR LLMs rely on next-token prediction, while diffusion LLMs rely on iterative denoising conditioned on noised latent states. Whether the diffusion training objective can effectively support long-context scaling was unknown prior to our investigation. Therefore, we intentionally isolate this factor by adopting NTK—the only DLLM-tested scaling method—to ensure a controlled comparison between **training-free** and **training-based** strategies.
>
> Finally, our choice does not imply that PI, YaRN, or other fine-grained AR-developed scaling methods are incompatible with diffusion models. Rather, these approaches introduce depth-dependent or frequency-modulated behaviors whose interactions with the diffusion denoising process and training objectives remain unexplored. Adapting AR-designed scaling mechanisms to diffusion LLms is itself an oen research problem, and we consider it a promising direction fo future work beyond the scope of this study.

---

> ### Author Response · Authors · 2025-11-21
>
> > Q1: Can you provide theoretical or empirical analysis showing that diffusion models actually learn relative positions in the range [-2T_train, 2T_train] during pre-training?
> >
>
> We thank the reviewer for raising the question. Below, we provide a clearer theoretical justification, based on the structure of RoPE, for why a diffusion model trained on sequences of length $T_{train}$ naturally learns relative positions within the range $[-(T_{train}-1),T_{train}-1].$
>
> **(1) RoPE makes attention scores depend on relative offsets $(j-i).$**
>
> Let the head dimension be $d$. We split the query and key into $d/2$ two-dimensional groups:
>
> $$
> \bar{q} = (q_{0},q_{1},q_{2},q_{3},...,q_{d-2},q_{d-1})^{T},\\
> \bar{k} = (k_{0},k_{1},k_{2},k_{3},...,k_{d-2},k_{d-1})^{T},\\
> $$
>
> The $m$-th 2D group is rotated with frequency $\theta_{m}$.
>
> The RoPE rotation for position $i$ is a block-diagonal matrix:
>
> $$
> R(i) = diag(R_{0}(i\theta_{0}),R_{1}(i\theta_{1}),...,R_{d/2-1}(i\theta_{d/2-1})),\\
> R_{m}(i\theta_{m}) = \begin{pmatrix} \cos(i\theta_{m}) & -\sin(i\theta_m) \\ \sin(i\theta_m) & \cos(i\theta_m)\end{pmatrix}.
> $$
>
> After position encoding:
>
> $$
> q_{i} = R(i)\bar{q}, \quad k_{j}=R(j)\bar{k}.
> $$
>
> The attention score between the $i$-th query and $j$-th key becones
>
> $$
> Attn_{ij}=q_{i}^{T}k_{j}=\bar{q}^{T}R(i))^{T}R(j)\bar{k}.
> $$
>
> For each 2D block,
>
> $$
> Attn_{ij}^{(m)}=(\bar{q}^{(m)})^{T}R_m((j-i)\theta_{m})\bar{k}^{(m)} = A_{m}\cos((j-i)\theta_{m})+B_{m}\sin((j-i)\theta_{m}),
> $$
>
> where the content-dependent coefficients
>
> $$
> A_m = q_{2m}k_{2m} + q_{2m+1}k_{2m+1},\quad B_{m} = q_{2m}k_{2m+1}-q_{2m+1}k_{2m}
> $$
>
> do not depend on positions.
>
> Thus, full attention is
>
> $$
> Attn_{ij} = \sum_{m=0}^{d/2-1}(A_{m}\cos((j-i)\theta_{m})+B_{m}\sin((j-i)\theta_{m})).
> $$
>
> This decomposition shows:
>
> - All positional effects enter **only** through $(j-i).$.
> - RoPE explicitly parameterizes attention as a function of the **relative offset**.
>
> **(2) What range of relative positions does the diffusion model see during training?**
>
> In pre-training of diffusion LLMs, all attention pairs satisfy
>
> $$
> i,j\in \set{0,...,T_{train}-1}.
> $$
>
> Therefore, the model is exposed to relative offsets:
>
> $$
> (j-i)\in [-(T_{train}-1), T_{train}-1].
> $$
>
> Because attention is a sum of terms over $(j-i)$, the diffusion model RoPE learns a function: $f_{RoPE}:(j-i) \mapsto Attn_{ij}$ directly over this offset domain.

---

### Official Review · Reviewer_BN3Q · 2025-11-03

**Soundness:** 3
**Presentation:** 3
**Contribution:** 2
**Rating:** 6
**Confidence:** 3

**Summary:**

This paper introduces UltraLLaDA, a diffusion LLM post-trained from LLaDA with a modified RoPE scaling method to extend the context length to 128K. UltraLLaDA proposes a diffusion-aware NTK/RoPE scaling method. The authors argue that standard NTK-aware RoPE scaling (as used in autoregressive long-context extension) is suboptimal for diffusion LLMs because diffusion models use bidirectional attention rather than causal attention. They propose a modified scaling that better reflects the positional distance statistics of bidirectional denoising. They then conduct long-context post-training on 64K length packed sequences, exploring several strategies for handling multiple concatenated documents: naive concatenation, EOD-token concatenation, and adaptive attention masking. The experimental results show that UltraLLaDA maintains near-perfect Needle-in-a-Haystack retrieval up to 128K and keeps perplexity stable out to 128K, whereas both the original LLaDA and the training-free method LongLLaDA collapse much earlier.

**Strengths:**

1. **Good practical significance for diffusion LLMs.**
   Ultra-long context capability in diffusion LLMs has not been well explored. This work provides a realistic recipe to reach 128K with stable retrieval and usable perplexity, which is a meaningful capability milestone for dLLMs.

2. **Comprehensive study of the proposed diffusion-aware NTK.**
   The paper presents a detailed analysis of the proposed diffusion-aware NTK, comparing it with LongLLaDA's baseline NTK, and shows the empirical improvement of diffusion-aware NTK in the training-free setting. This analysis is convincing.

3. **Comprehensive experimental results and ablations.**
   This work conducts experiments on various long-context tasks, including Needle-in-a-Haystack, LongBench, and RULER, for all baselines (LLaDA, LongLLaDA) and UltraLLaDA trained with different sentence packing strategies. This thorough comparison supports the proposed method.

4. **Clear writing.**
   The paper is clearly written and easy to follow.

**Weaknesses:**

1. **Core novelty feels incremental.**
   The main contribution of this work is diffusion-aware NTK. While the motivation (bidirectional vs. causal attention) is reasonable, the scaling rule itself shows only a moderate improvement over LongLLaDA’s baseline scaling when evaluated without post-training. From Table 4 and Table 5, even with post-training, the improvement for 4K–16K context lengths still seems small. The three long-context post-training sentence-packing strategies explored in this paper are also existing methods in autoregressive post-training. Applying and comparing them in the diffusion LLM setting is empirically valuable but not conceptually new.

2. **Limited analysis of non-retrieval reasoning at longer context lengths.**
   Most ultra-long-context evaluations are on NIAH and perplexity stability. The paper does evaluate on LongBench with 16K context length and RULER at 32K, but there is less evidence for complex multi-document synthesis or instruction following at 64K–128K.

3. **Lack of evaluation on more models.**
   The paper only conducts experiments on LLaDA model. There are multiple kinds of diffusion LLMs: models trained from scratch in diffusion style like LLaDA, models converted from AR like Dream, and block-diffusion-style models. This work should apply the proposed method to more diffusion LLMs to demonstrate generalization.

**Questions:**

1. Could you evaluate UltraLLaDA on more kinds of tasks beyond PPL and NIAH at longer context lengths? Are there any failure cases at 128K context length (because the current NIAH evaluation results seem perfect)?

2. Could you apply the proposed method to models like Dream (which is trained from AR model) and SDAR-series models (which are also trained from AR models with block diffusion style)?

---

> ### Author Response · Authors · 2025-11-21
>
> > W1: The main contribution of this work is diffusion-aware NTK. While the motivation (bidirectional vs. causal attention) is reasonable, the scaling rule itself shows only a moderate improvement over LongLLaDA’s baseline scaling when evaluated without post-training. From Table 4 and Table 5, even with post-training, the improvement for 4K–16K context lengths still seems small. The three long-context post-training sentence-packing strategies explored in this paper are also existing methods in autoregressive post-training. Applying and comparing them in the diffusion LLM setting is empirically valuable but not conceptually new.
> >
>
> We thank the reviewer for raising the concern. We conducted additional long-context experiments to directly compare diffusion-aware NTK with the baseline NTK under identical post-training settings.
>
> **(1) Effectiveness of Diffu-aware NTK.**
>
> To evaluate the effectiveness of diffu-aware NTK under the training settings, we extended our evaluation on both scaling methods to longer contexts beyond those reported in the paper:
>
> - RULER with 48K context length.
> - LongBengch-v2 with 64K context length.
>
> Across all tasks, diffusion-aware NTK outperforms baseline NTK: (LB- denotes LongBench-v2 tasks.)
>
> |  | LB-Easy | LB-Hard | LB-Short | LB-Medium | LB-Long | RULER |
> | --- | --- | --- | --- | --- | --- | --- |
> | Diffu-aware NTK | 35.94 | 28.94 | 37.22 | 27.91 | 29.63 | 56.94 |
> | Baseline NTK | 33.33 | 27.97 | 35.56 | 27.44 | 25.93 | 53.75 |
>
> These results indicate that incorporating diffusion-specific relative position spans leads to more stable long-context behavior than directly applying AR-centric NTK scaling.
>
> **Evaluation Settings.**
>
> - RULER. We set output length, block length, and denoising step all to 64;
> - LongBench-v2. We set output length, block length, and denoising step all to 4. The prompt template is: 'Please read the following text and answer the question below.\n\n<text>\n{context}\n</text>\n\nWhat is the correct answer to this question: {question}\nChoices:\n(A) {choice_A}\n(B) {choice_B}\n(C) {choice_C}\n(D) {choice_D}\n\nAnswer:'
>
> **(2) Packing strategies.**
>
> Our contribution in this section is to summarize the practical behavior of different packing strategies, specifically *in the diffusion-LLM setting*, where global bidirectional attention makes cross-document interference more pronounced than in AR models. Our empirical finding is simple: **EOD concatenation is preferable at short to medium lengths**, while **Adaptive Masking is consistently more robust at very long lengths**. Both outperform naive concatenation. We will make this conclusion more explicit in the revision.

---

> ### Author Response · Authors · 2025-11-21
>
> > W2: Most ultra-long-context evaluations are on NIAH and perplexity stability. The paper does evaluate on LongBench with 16K context length and RULER at 32K, but there is less evidence for complex multi-document synthesis or instruction following at 64K–128K.
> >
>
> > Q1: Could you evaluate UltraLLaDA on more kinds of tasks beyond PPL and NIAH at longer context lengths? Are there any failure cases at 128K context length (because the current NIAH evaluation results seem perfect)?
> >
>
> We thank the reviewer for pointing out the questions. To address it, we expanded our evaluation to include more complex comprehension and reasoning tasks at longer context lengths.
>
> **(1) Additional long-context evaluations beyond PPL and simple NIAH.**
>
> We conducted new experiments on:
>
> - RULER with 48K context length.
> - LongBengch-v2 with 64K and 96K context length.
>
> UltraLLaDA’s performance is summarized below:
>
> | UltraLLaDA | AGG | QA | VT | NIAH |
> | --- | --- | --- | --- | --- |
> | ruler-48K | 25.12 | 24.5 | 95.2 | 68.22 |
>
> | UltraLLaDA | Easy | Hard | Short | Medium | Long |
> | --- | --- | --- | --- | --- | --- |
> | long bench-v2-64K | 35.94 | 28.94 | 37.22 | 27.91 | 29.63 |
> | longbench-v2-96K | 38.54 | 27.97 | 37.22 | 27.77 | 27.78 |
>
> The evaluation configuration and prompt template are the same as the response to W1.
>
> These benchmarks contain diverse tasks requiring reasoning and understanding. The results show that UltraLLaDA maintains performance when scaling to longer contexts, demonstrating its robustness beyond information-retrieval and perplexity-based metrics.
>
> **(2) Instruction-following.**
>
> Our study evaluates the **base model**, not an instruction-tuned variant. Since instruction-following capability is not the focus of the base model, we do not include dedicated instruction-following evaluations in this work.
>
> **(3) Failure cases.**
>
> Although UltraLLaDA achieves 100% accuracy on the simple NIAH-128K setup, we do see meaningful failures on harder RULER tasks. Typical failure modes include:
>
> - Generating only newline characters (`"\n"`),
> - Repeating the question instead of retrieving the relevant sentence.
>
> We will highlight these failure cases in Section 4.1, to give a balanced picture of UltraLLaDA’s strengths and limitations at extreme context lengths.
>
> Overall, we hope these new experiments and analyses address the reviewer’s concerns by demonstrating the effectiveness of UltraLLaDA across a broader range of long-context tasks, while also acknowledging realistic failure cases.

---

> ### Author Response · Authors · 2025-11-21
>
> > W3: The paper only conducts experiments on LLaDA model. There are multiple kinds of diffusion LLMs: models trained from scratch in diffusion style like LLaDA, models converted from AR like Dream, and block-diffusion-style models. This work should apply the proposed method to more diffusion LLMs to demonstrate generalization.
> >
>
> > Q2: Could you apply the proposed method to models like Dream (which is trained from AR model) and SDAR-series models (which are also trained from AR models with block diffusion style)?
> >
>
> We thank the reviewer for highlighting the concern. To address this, we extended our study beyond LLaDA and applied our training-based scaling approach to the Dream model (Dream-v0-Base-7B), a representative diffusion LLM converted from an autoregressive model. As for the SDAR-series models, only their Chat variants are publicly released while the corresponding Base models are not available, so we do not include them in our evaluation.
>
> **(1) Applying our training-based method to Dream.**
>
> **Setup.**
>
> Dream has a native **2K** context and retains an AR-style shifted-label architecture, which is structurally different from LLaDA’s unshifted diffusion training. We extend Dream from 2K to 32K using our training-based pipeline and compare:
>
> 1. **Dream**: original 2K model.
> 2. **Dream-NTK**: direct NTK scaling (LongLLaDA-style) without any training.
> 3. **Dream-finetune (32K Dream)**: Dream post-trained on 32K data with our method.
>
> We evaluate on **NIAH-32K** and **LongBench-v2-short (≤32K)**:
>
> |  | N-4K | N-8K | N-16K | N-24K | N-32K | long bench-v2-short(≤32K) |
> | --- | --- | --- | --- | --- | --- | --- |
> | Dream | 100 | 85.2 | 36.7 | 23.7 | 19.3 | 30.00 |
> | Dream-ntk | 92.1 | 59.8 | 19.8 | 7.7 | 4.1 | 27.78 |
> | Dream-finetune | 100 | 100 | 100 | 100 | 100 | 31.11 |
>
> （The evaluation configuration and prompt template are the same as the response to W1.）
>
> **(2) Key empirical finding.**
>
> - Directly applying **NTK-scale (LongLLaDA’s method)** to the original, untrained Dream model **degrades** long-context performance—below even the original 2K Dream baseline.
> - In contrast, with our **training-based approach**, the 32K Dream model achieves **the best overall performance** among the three variants: Original Dream, NTK-scaled Dream and our 32K-Dream.
>
> This demonstrates that the **training-based strategy generalizes successfully** to diffusion LLMs that originate from AR models and use different training objectives.

---

### Author Response · Authors · 2025-12-03
**Summary**

Dear Area Chair,

We hope this message finds you well, and we sincerely appreciate your time and effort in handling our submission. As the discussion period is coming to an end, we would like to offer a brief summary to ensure you have the full context of our work when making the final decision.

**Summary of Main Contribution**.

UlrtaLLaDA is a post-training recipe for extending diffusion LLMs from 4K to 128K context.

Existing long-context methods are exclusively developed for autoregressive LLMs, while diffusion LLMs differ fundamentally in both training objective (mask token denoising vs. next-token prediction) and attention pattern (global bidirectional vs. causal). Whether diffusion LLMs can be stably scaled to ultra-long context via training-based methods was essentially unknown. UltraLLaDA addresses this gap with two main contributions:

- Diffusion-aware NTK RoPE scaling. We adapt NTK-aware RoPE to diffusion LLMs by accounting for the fact that diffusion models employ global bidirectional attention, which induces different positional-distance statistics compared to autoregressive training.
- We systematically compare naive concatenation, EOD-token concatenation, and Adaptive Masking under diffusion training, where cross-document interference is severe due to global bidirectional attention. Our findings also give practical guidance.

Empirically, UltraLLaDA achieves 100% simple NIAH accuracy at 128K, maintains stable perplexity out to 128K, and substantially outperforms both base LLaDA and the training-free LongLLaDA baseline on LongBench and RULER.

**Regarding Reviews**.

All four reviewers (BN3Q, eBLu, ev1L, S1Du) assign an overall rating of 6, with confidence levels between 3 and 5. Across the reviews, they acknowledge that:

- The problem of long-context scaling for diffusion LLMs is novel and important.
- The proposed diffusion-aware NTK is intuitive and practically effective.
- The paper is clearly written and supported by comprehensive experiments and meaningful ablations.

Their main concerns center on evaluation breadth, generalization of our training-based method, and theoretical clarification of the proposed approach.

During the discussion period, we conducted sustantial new experiments and provided detailed theoretical clarifications targeting these concerns.

**New Experiments During Discussion Period**.

Below, we summarize the new experiments that directly address the reviewers’ primary concerns. These were completed after the original submission and documented in our discussion posts.

- Longer-Context Reasoning (Reviewers BN3Q, eBLu, S1Du). We extended evaluation *beyond PPL and NIAH* to more complex long-context tasks, including RULER at 48K and LongBench-v2 at 64K and 96K.
- Training-Based Diffusion-Aware NTK vs Baseline NTK at Long Context (Reviewers BN3Q). We conducted a direct comparison between Diffusion-aware NTK and baseline NTK under identical post-training settings, evaluating on RULER at 48K and LongBench-v2 at 64K.
- Generalization to Dream (Diffusion LLM Converted from AR) (Reviewers BN3Q, eBLu). To test generalization, we applied our training-based method to Dream-v0-Base-7B (converted from an AR model via shifted-label architecture). We compared Dream, Dream-NTK (direct LongLLaDA-style NTK), and Dream-finetune (our method).
- Critical-Dimension Sweep for Diffusion-Aware NTK (Reviewer S1Du). We extended the analysis of the critical dimension by sweeping $d_{crit} \in \set{69, 70, 71, 73, 76, 79}$, evaluated on LongBench-v2-short (≤32K) and NIAH (8K~32K).

**New Clarifications During Discussion Period**.

Below summarizes the main clarifications provided during the discussion.

- Why NTK (and not YaRN/PI) as the Main Baseline (Reviewers eBLu, S1Du). LongLLaDA with NTK is currently the only empirically validated long-context scaling method for diffusion LLMs. Our central scientific question is: “Can training-based context extension be effective for diffusion LLMs at al*l, given their different objectives and attention structure?”* To isolate this question, we deliberately fix the RoPE scaling family to NTK and compare training-free versus training-based strategies.
- Theoretical Justification of the “2× Range” for Relative Positions (Reviwers BN3Q, ev1L, S1Du). We added a  theoretical derivation based on RoPE’s structure to justify the effective doubling of the relative-position range.
- Masking Strategies for Diffusion LLMs (Reviewers BN3Q, eBLu, ev1L). We clarified our conclusions on different masking strategies under global biderectinal attention.
- Short-Context Degtadation and Context Length Limit (Reviewers S1Du, eBLu). We explicitly acknowledged that short-context degradation after long-context fine-tuning—also shown in our appendix. We further discussed the limits to context length.

We believe these addtional experiments and clarifications comprehensively respond to the core concerns raised by the reviewers.

Best regards,

The Authors of Submission 5383

---

### Meta-Review · Area_Chair_SGcE · 2026-01-07

**Summary:**

Reviewers agreed that the paper is addressing a previously unexplored problem in diffusion LMs. They also liked the overall empirical results. Major concerns raised:

1. The core contribution (diffusion-aware NTK scaling and masking strategies) was conceptually incremental relative to prior long-context work in AR models, even if empirically valuable in the diffusion setting.
2. Weak evaluation: only perplexity and NIAH were reported, with limited evidence for more complex long-context reasoning or synthesis at very long lengths.
3. Unclear whether the approach would generalize beyond Llada to other models or architectures (like Dream)
4. There were also some concerns about the narrow baseline set and the short-context degradation observed after long-context post-training.

While not all of these major concerns are addressed, and authors have acknowledged certain limitations, I think the paper attempts to solve previously unexplored problem (and should count as novel work). While not perfectly executed, the issues are not limiting and can be addressed in future work.

**Reviewer Concerns:**

1. The authors added new experiments on RULER  and LongBench-v2.
2. The additional experiments on Dream 7b demonstrate that their method can work on other diffusion models converted from AR architectures.
3. The authors acknowledged failure cases at long context and discussed the rationale for the 128K limit.

Outstanding:
1. The paper still does not include comparisons to YaRN/PI-style scaling, though the authors provided a reasonable justification for focusing on NTK given the diffusion-specific context.
2. While acknowledged and explained, mitigation strategies are deferred to future work rather than demonstrated.

**Reviewer Scores:**

All reviewers already lean accept, some also further increased their scores (before participation was closed).

---

### Decision · Program_Chairs · 2026-01-26

Accept (Poster)